# Probing non-equilibrium topological order on a quantum processor

M. Will[1,2,8], T. A. Cochran[3,8], E. Rosenberg[4], B. Jobst[1,2], N. M. Eassa[4,5], P. Roushan[4✉], M. Knap[1,2✉], A. Gammon-Smith[6,7✉] & F. Pollmann[1,2✉]

Out-of-equilibrium phases in many-body systems constitute a new paradigm in quantum matter—they exhibit dynamical properties that may otherwise be forbidden by equilibrium thermodynamics. Among these non-equilibrium phases are periodically driven (Floquet) systems[1–5], which are generically difficult to simulate classically because of their high entanglement. Here we realize a Floquet topologically ordered state theoretically proposed in ref. 6, on an array of superconducting qubits. We image the characteristic dynamics of its chiral edge modes and characterize its emergent anyonic excitations. Devising an interferometric algorithm allows us to introduce and measure a bulk topological invariant to probe the dynamical transmutation of anyons for system sizes up to 58 qubits. Our work demonstrates that quantum processors can provide key insights into the thus-far largely unexplored landscape of highly entangled non-equilibrium phases of matter.

Quantum many-body systems have a rich landscape of equilibrium phases of matter. Among them are symmetry-breaking phases and topological phases[7–9]. Although the former are described by local order parameters, the latter are instead characterized by non-local entanglement[10,11] and the emergence of fractionalized anyonic excitations[12–14]. The non-local nature of states with topological order is the underlying principle leveraged in many quantum error correcting codes[15], in which quantum information is encoded in locally indistinguishable ground states that are robust to local perturbations. Recent experimental advances allow the realization of these topological states on quantum processors, through scalable quantum circuits[16–20].

It has been predicted early on that time-periodic driving can lead to the emergence of topologically protected non-equilibrium phases of matter that are fundamentally distinct from those in thermal equilibrium[21]. A recently proposed example in two-dimensional systems shows that periodically driving non-interacting particles can support robust chiral edge states, even when the Chern numbers of all bulk bands are zero—a property forbidden in static band structures[1–3,22,23]. Moreover, discrete time crystals[4,5,24–30] can be realized in driven systems, which, crucially, cannot be stabilized in any equilibrium scenario[31–33]. Their prediction and subsequent realization[26–30] brought spatio-temporal ordering to focus and motivated further exploration of symmetry-protected non-equilibrium phases of matter[34–36] as well as topologically ordered time crystals[37,38].

This leads to a central question as to whether new types of topological order can be realized away from equilibrium and by time-periodic driving. Theoretical studies propose the Floquet Kitaev model as a paragon[6,39]—a periodically driven version of Kitaev's honeycomb model[40]. Under strong driving, this system has been predicted to exhibit Floquet topological order (FTO)[6] and time-vortex excitations[41].

Here, we implement efficient quantum circuits in a two-dimensional lattice of superconducting qubits and probe two key non-equilibrium signatures of the Floquet topologically ordered phase. The first is the presence of topologically protected chiral edge modes hosting non-Abelian Majorana modes. Importantly, these edge modes occur with zero Chern number for the bulk bands[6]. This is in sharp contrast to the equilibrium setting, in which a non-zero integer value of this topological invariant is a requisite for the existence of chiral modes. The second distinguishing signature is the non-equilibrium topological order in the bulk of the system. Unique to this Floquet setting are two distinct anyon types transmuting between each other, alternating with twice the period of the drive. Quantum processors allow us to adjust parameters away from the analytically tractable regimes of the Floquet Kitaev model and examine the stability of FTO for different parameter regimes. These are regimes in which, due to the rapid generation of entanglement, the simulation abilities of classical computers are strongly limited. Quantum processors further allow us to design protocols for measuring observables that unambiguously relate to the underlying physics.

## Realizing the Floquet Kitaev model

The Floquet Kitaev model, introduced in ref. 6, is a periodically driven analogue of Kitaev's honeycomb model[40]. In much the same way that Kitaev's honeycomb model is the archetype for quantum spin liquids in equilibrium, the Floquet Kitaev model provides an exactly solvable model for FTO. It consists of spin-1/2 degrees of freedom arranged on the vertices of a honeycomb lattice as shown in Fig. 1a. The three types

[1]TUM School of Natural Sciences, Physics Department, Technical University of Munich, Garching, Germany. [2]Munich Center for Quantum Science and Technology (MCQST), Munich, Germany. [3]Department of Physics, Princeton University, Princeton, NJ, USA. [4]Google Research, Mountain View, CA, USA. [5]Department of Physics and Astronomy, Purdue University, West Lafayette, IN, USA. [6]School of Physics and Astronomy, University of Nottingham, Nottingham, UK. [7]Centre for the Mathematics and Theoretical Physics of Quantum Non-Equilibrium Systems, University of Nottingham, Nottingham, UK. [8]These authors contributed equally: M. Will, T. A. Cochran. ✉e-mail: pedramr@google.com; michael.knap@ph.tum.de; Adam.Gammon-Smith@nottingham.ac.uk; frank.pollmann@tum.de

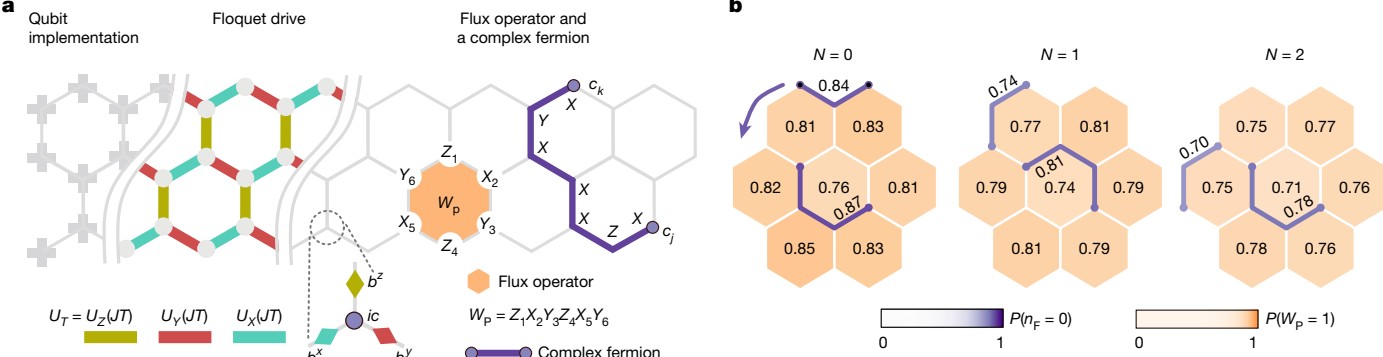

**Fig. 1 | Realization of the Floquet Kitaev model. a**, The hexagonal lattice can readily be embedded on the square lattice geometry of the superconducting qubits. Stroboscopic time evolution realizes the Floquet Kitaev model, where coloured bonds correspond to the direction-dependent couplings. On each site, the qubit can be mapped to four Majorana operators $b^x$, $b^y$, $b^z$ and $c$. A plaquette operator $W_P$, which is a Pauli string around the hexagons of the lattice, is highlighted. Pairing two $c$-Majoranas together results in a complex fermion shown in purple, and its occupation number operator is given by the indicated Pauli string. **b**, Measurements for fluxes and paired emergent Majorana modes with occupation $n_F = 0$, during the stroboscopic time evolution. Here we specifically track one fermion occupation along the edge and one in the bulk (other fermions are not shown for clarity) for up to $N = 2$ cycles. Data taken with dynamical decoupling and randomized compiling for CZ gates ($N_{twirling} = 20$, $N_{shots} = 10^6$, 26 qubits).

of bonds are labelled by $X$, $Y$ and $Z$, indicating the type of direction-dependent Ising coupling applied on that bond. We implement the stroboscopic evolution given by

$$U_T = U_Z(JT)\ U_Y(JT)\ U_X(JT) \tag{1}$$

where $U_\alpha(JT) = \exp\left\{-i\frac{\pi}{4}JT\sum_{\langle j,k\rangle_\alpha}\alpha_j\alpha_k\right\}$, with Pauli operators $\alpha \in \{X, Y, Z\}$. The sums run over all neighbouring $j$, $k$ spin pairs of the corresponding bond type. The dynamics are controlled by $JT$, where $J$ is the coupling parameter and $T$ is the driving period. We implement the $U_\alpha$ driving terms with single-qubit rotations and two-qubit C-PHASE gates (details in the Methods). Depending on the strength of the driving, the model is in one of several non-equilibrium phases[6,39]. The FTO phase, as a distinct non-equilibrium phase of matter, is predicted to exist close to $JT = 1$ even away from the integrable case[6]. The Floquet Kitaev model is well understood because of a Majorana representation of the spins, which shows the free-fermion solvability of the model[6,39]. On each site, four Majorana operators $\{c, b^x, b^y, b^z\}$ are defined as shown in Fig. 1a, such that the Pauli operators are $\alpha_j = ic_jb_j^\alpha$. This representation enlarges the Hilbert space, and the physical states $|\psi\rangle$ are those with $c_jb_j^xb_j^yb_j^z|\psi\rangle = |\psi\rangle$ for all $j$. The Floquet Kitaev model has a set of conserved quantities $u_{jk} = ib_j^\alpha b_k^\alpha$, where $\alpha$ is the Pauli operator associated with the bond $\langle j, k\rangle$. Products of the operators $u_{jk}$ around the hexagonal plaquettes $P$ as shown in Fig. 1a correspond to $\mathbb{Z}_2$ gauge-invariant conserved quantities

$$W_P = \prod_{\langle j,k\rangle\in\bigcirc} u_{jk}, \tag{2}$$

where we refer to $W_P = +1$ as flux-free. Using the Majorana representation, the driving terms are then given by $U_\alpha(JT) = \exp\left\{-JT\frac{\pi}{4}\sum_{\langle j,k\rangle_\alpha}u_{jk}c_jc_k\right\}$, which are quadratic in the $c_i$-Majorana fermion operators. These terms correspond to hopping operators for the $c$-fermions and perform a Majorana swap when $JT = 1$. To monitor the dynamics of the emergent Majorana excitations, we define measurable density operators of a complex fermion $\psi$ resulting from pairing two $c$-Majoranas at sites $j$ and $k$,

$$n_F(j, k) = (i\phi_{jk}c_jc_k + 1)/2, \quad \text{with} \quad \phi_{jk} = \prod_{\langle l,m\rangle\in\Gamma}u_{lm}, \tag{3}$$

and $\Gamma$ is a path connecting those sites. By including the string $\phi_{jk}$, this density operator can be equivalently written as a Pauli string and thus be directly measured on the quantum processor (see Methods for details).

By stroboscopic measurements, we visualize the dynamics of the paired Majoranas in the FTO phase. For this, we first implement a unitary circuit $U_{FF}$ to prepare the system in a flux-free (FF) state,

$$|\Psi_{FF}\rangle = U_{FF}|0\rangle^{\otimes N}, \quad \text{with} \quad W_P|\Psi_{FF}\rangle = |\Psi_{FF}\rangle\ \forall\ P. \tag{4}$$

This is achieved by noting the relationship to the ground states of the toric code, which can be prepared efficiently with unitary circuits with a depth linear in the width of the system[16,42]. The preparation also fixes a unique pairing of Majoranas into complex fermions with $n_F = 0$ on neighbouring sites on the $z$-bonds (not shown here; see Methods). Using a sequence of bond operators of the form $\exp\left\{-\frac{\pi}{4}u_{jk}c_jc_k\right\}$, the Majoranas are swapped until the state $|\Psi(0)\rangle$ with the desired pairing of Majoranas is reached (which can be non-local). The state $|\Psi(0)\rangle$ is then the initial state for the subsequent evolution with the Floquet unitary $U_T$. Figure 1b shows the measured fluxes $W_P$ and the dynamics of two selected densities $n_F(j, k)$ of paired Majoranas following the Floquet evolution with $JT = 1$. The bulk Majoranas form closed orbits of period two, and the chiral motion of the fermions along the edge is visible—resembling the (skipping) cyclotron orbits in the semi-classical picture of quantum Hall physics. The experimental data demonstrate an approximate conservation of the plaquette fluxes $W_P$ and densities up to noise that accumulates with circuit depth.

## Chiral Majorana edge mode interferometry

We next show the non-trivial exchange statistics of the Majorana edge modes. The exchange of a pair of Majorana excitations results in the accumulation of a phase that depends on the fermion parity of the pair. More precisely, if the pair of Majorana excitations corresponds to an occupied fermionic state $\langle n_F\rangle = 1$, then exchange will lead to a phase factor of $e^{-i\pi/4}$, whereas an unoccupied fermion $\langle n_F\rangle = 0$ gives $e^{i\pi/4}$. To extract this topological phase information experimentally, we first prepare $|\Psi_{FF}\rangle$ and then modify it to $|\Psi(0)\rangle$, which has a Majorana pair stretched across the system as shown in Fig. 2a. For the chosen setup, the occupancy can be changed by acting with a Pauli $X$ on qubit 0. The Floquet driving drags the modes around the edge of the system until the Majoranas are exchanged after a system-size-dependent number of time steps and returned to an equivalent configuration (Fig. 2b).

Similar to a Hadamard test using an ancilla qubit, we devised a pulse sequence that allows us to measure the—otherwise global—relative phase accumulation by the quantum states. The key element is a controlled operation, which enables the parallel evolution of an occupied

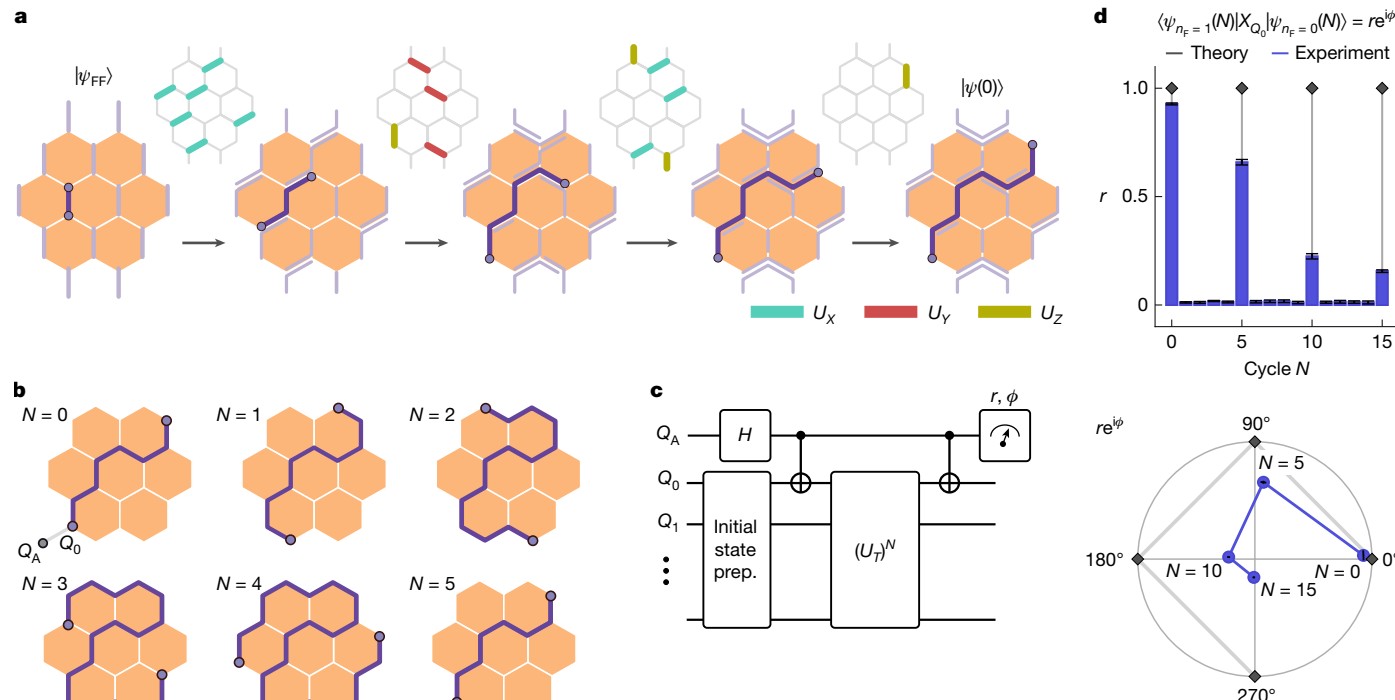

**Fig. 2 | Dynamical Majorana edge mode interferometry. a**, Gate sequence on top of the flux-free state $|\psi_{FF}\rangle$ to pair Majorana modes on opposite sides of the system. **b**, Transformation of the stretched, complex fermion under Floquet driving. For clarity, other fermions are not shown. **c**, Circuit for the Hadamard test to measure the overlap between evolved states with the fermion occupied and unoccupied to extract the relative phase of $e^{i\pi/2}$. **d**, Radial component of the overlap, comparing data taken on the device with theory. Polar plot of the complex value of the overlap shown at multiples of $N = 5$ cycles. Error bars correspond to $2\sigma$ obtained by jackknife resampling (for details, see Methods). Data taken with dynamical decoupling, randomized compiling and post-selection of the flux through each plaquette of the outer ring, see Methods ($N_{\text{twirling}} = 20$, $N_{\text{shots}} = 5 \times 10^5$, 27 qubits).

and an unoccupied Majorana pair and maps the relative phase of the two states to the ancilla qubit ($Q_A$) (Fig. 2c). We compare the two occupations for the non-local pair of Majoranas, which enables us to read off the amplitude $r$ and phase $\phi$ of this overlap. The overlap shows revivals with period $N = 5$, shown in Fig. 2d, matching the period of exchange for the highlighted pair of Majoranas for the considered geometry in Fig. 2b. Compared with the theoretically expected value $r = 1$ for the amplitudes of the revivals, we observe decay due to errors and decoherence on the processor. For intermediate time steps, the final controlled operation does not simply flip the fermion occupation, and thus creates an orthogonal state for which the overlap is zero as expected. By measuring the real and imaginary parts of the overlap, we observe a relative phase of $e^{i\pi/2}$ at multiples of the period of $N = 5$, which demonstrates that the Majorana modes at the edge are dynamically exchanged. Although the measured amplitude shows a notable decay, we find the measured complex phase to be remarkably stable.

## Spectroscopy of edge modes

So far, we have focused on the analytically solvable fixed point circuit by setting $JT = 1$. The FTO, however, represents an extended non-equilibrium phase of matter that remains stable in the thermodynamic limit in the non-interacting case. It is also expected to be protected over long time scales in the presence of interactions, provided there is sufficient disorder to stabilize a pre-thermal many-body localized bulk while retaining FTO[6]. Now we probe the robustness of the edge modes and image the spectrum by measuring the two-time Majorana spectral function along the edge of the system. The Fourier transform of this function shows the momentum- and quasi-energy-resolved spectrum of the chiral edge mode. This spectral function is not directly measurable as it involves individual Majorana operators at a given time, which are not accessible. However, we can augment the system by an additional qubit (Fig. 3a, protruding bond) that is not participating in the time evolution but instead is used to pair with the desired Majorana mode. The information of the edge mode can then be equivalently measured with unequal-time correlators of Pauli strings

$$C(j, N) \propto \langle\psi_0|P_j(N)P_0(0)|\psi_0\rangle, \tag{5}$$

where $P_j(N)$ is the corresponding Pauli string tracing the edge fermions evolved under $N$ Floquet cycles. This quantity would correspond to a four-point Majorana correlator. However, the modes on the undriven additional qubit cancel, leaving the sought-after equivalent two-point unequal-time Majorana correlator (Methods). Similar to the last pulse sequence, we use a Hadamard test to measure this unequal-time Pauli correlator, as shown in Fig. 3c. To additionally probe the robustness of the edge modes, we move away from the fixed point $JT = 1$ in two ways. We first detune the driving from this point by considering $JT \neq 1$. We also modify the drive sequence by adding a disordered $Z$-field as a fourth step in the stroboscopic evolution shown in Fig. 3b and given by

$$U_{\text{disorder}} = e^{-iJT\pi/4 \sum_i h_i Z_i} \tag{6}$$

with $h_i \in [-\Delta, \Delta]$. This term explicitly breaks the integrability of the model by introducing dynamics to the fluxes $W_P$.

In the FTO phase, $JT = 1$ and $JT = 0.9$, the chiral edge mode is visible in the energy-momentum-resolved spectral function (Fig. 3e, bottom). Tuning away from the fixed point has the effect of a very weak broadening corresponding to a decay of the mode, which is best visible in the real-space data (Fig. 3e, top). This decay is expected because we are measuring the spectral function on the edge sites only, and the edge modes are not solely supported on these sites away from $JT = 1$. The observed decay at $JT = 1$ is due to the combination of coherent errors and decoherence in the device, which affects all other data sets

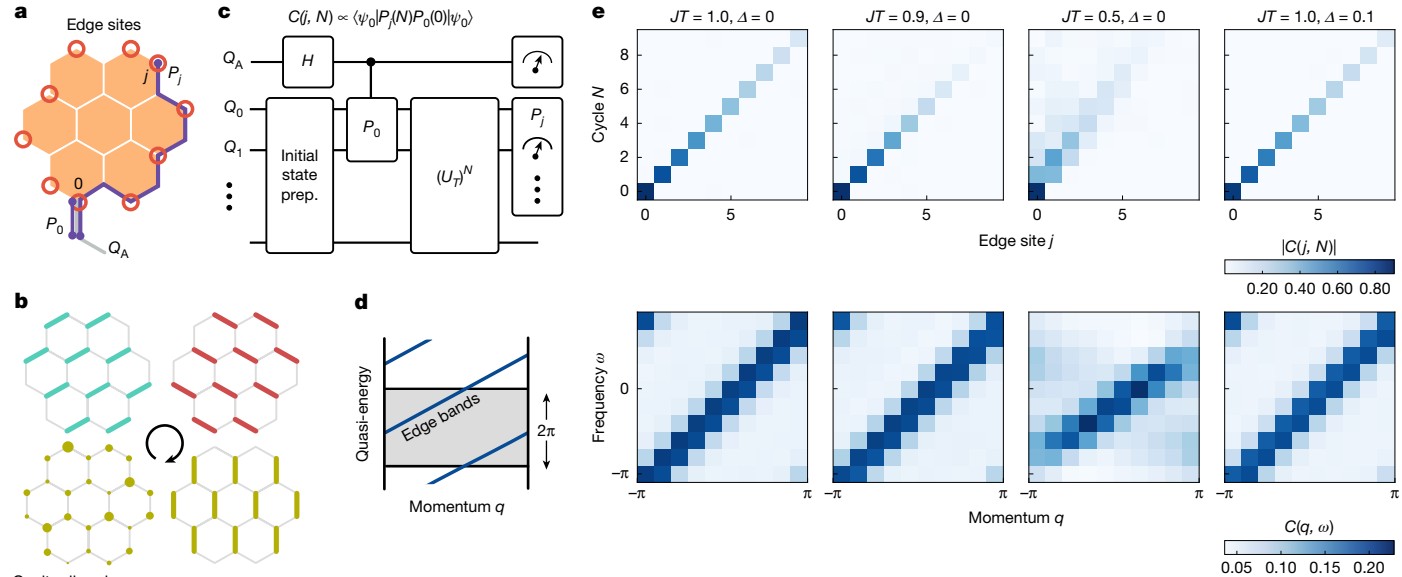

**Fig. 3 | Majorana edge mode spectrum. a**, The hexagonal lattice with edge sites shown in red. We also indicate the Pauli strings corresponding to the fermion occupation for two of the sites on the edge. **b**, Addition of a fourth time step with a random onsite $Z$-field that breaks integrability is given by $U_{\text{disorder}} = e^{-jT\pi/4 \sum_i h_i Z_i}$ with $h_i \in [-\Delta, \Delta]$. **c**, Schematic of the circuit to measure the unequal-time Pauli-string correlator, which is equivalent to the real-space Majorana spectral function, see text for more details. **d**, Exemplary sketch of spectrum in FTO phase. Edge states are shown in blue. **e**, The first row shows the spectral function in real space for $\Delta = 0$ and varying values of $JT$ (first three columns), and for $JT = 1$, $\Delta = 0.1$ (last column), in which the data are averaged over $N_{\text{av}} = 20$ disorder configurations. The second row shows the Fourier-transformed data. Data are taken with dynamical decoupling, randomized compiling for CZ gates and post-selection on the middle plaquette in experiments without a disordered field (first three columns), see Methods ($N_{\text{twirling}} = 20$, $N_{\text{shots}} = 5 \times 10^5$, 27 qubits).

as well. Importantly, we observe the non-trivial winding of the mode in quasi-energy that is unique to this non-equilibrium phase. Furthermore, moving to the non-Abelian Kitaev phase, $JT = 0.5$, we observe a qualitative difference in the dispersion of the chiral mode. In particular, there is no clear winding in quasi-energy, supporting that this regime is connected to the equilibrium phase. This is also visible in the real-space data, which shows how the mode quickly dissolves into the bulk. The introduction of the disordered field has a remarkably weak effect on the chiral mode for the experimentally accessible time scales. Although disorder in the interacting model is not expected to result in many-body localization in the thermodynamic limit[43], we anticipate the existence of an extended pre-thermal regime, during which the phase remains protected from heating[44–48].

## Bulk non-equilibrium topological order

The second distinguishing signature of the FTO phase is the bulk non-equilibrium topological order characterized by the transmutation of anyon types[6,39]. We use this unique phenomenology of the driven system to introduce an order parameter for the FTO phase. This order parameter is a topological invariant that oscillates between +1 and −1 in the FTO phase with twice the driving period and is constant in the non-Abelian Kitaev phase. In the bulk, the FTO phase is characterized by $\mathbb{Z}_2$ topological order, and the associated $e$, $m$ and $\psi = e \times m$ anyons[6]. The $m$ anyons correspond to flux defects ($W_P = -1$), $\psi$ is an occupied complex fermion ($n_F = 1$) introduced above, and the $e$ anyon is then the combination of a flux defect and an occupied fermion on a given plaquette (Fig. 4a). As the fermion parity $P_F = (-1)^{n_F} = \pm 1$ time evolves at the fixed point $JT = 1$ as

$$U_T^\dagger P_F U_T = W_P P_F, \tag{7}$$

the fermion occupation remains constant during time evolution on a flux-free plaquette with $W_P = 1$. However, if the flux is $W_P = -1$, the

fermion occupation will flip after each driving cycle, resulting in a periodicity of $N = 2$.

This transmutation of $e \leftrightarrow m$ anyons is a key characteristic of this non-equilibrium topological order, and can thus be referred to as an anyonic time crystal[6,37,38]. To probe this transmutation, we create an $e$ anyon in the central plaquette of the system and move the paired $e$ anyon to the boundary of the system, as shown in Fig. 4b. After evolving the system under the Floquet driving, we measure an electric loop operator $O$ encircling this central plaquette, shown in Fig. 4a (green). This loop operator corresponds to creating a pair of electric anyons, dragging one around the loop, and then annihilating the pair, and can be measured as a Pauli expectation value. To define an invariant, we compute the ratio of this quantity and the loop measurement without anyon at the centre, in the spirit of the Fredenhagen–Marcu order parameter[49],

$$\eta(N) = \frac{\langle \Psi_e | O(N) | \Psi_e \rangle}{\langle \Psi_0 | O(N) | \Psi_0 \rangle}, \tag{8}$$

where $O(N) = (U_T^\dagger)^N O (U_T)^N$. In the thermodynamic limit, and the limit of large loop diameter, this defines a bulk invariant for the phase that oscillates between +1 ($e$ anyon) and −1 ($m$ anyon) in the FTO phase.

Experimentally realizing a phase requires showing that its signatures survive over a pre-thermal time scale when perturbed away from the fine-tuned points in parameter space, that is, away from $JT = 1$, in which the drive sequence corresponds to a Clifford circuit. Probing a system of 58 qubits, the oscillations with period $N = 2$ are seen in the measured order parameter $\eta(N)$ for different values of $JT$ in the FTO phase (Fig. 4c). As $\eta(N)$ is estimated from a finite number of measurements, we regularize the denominator to account for shot noise (Methods). By contrast, in the non-Abelian Kitaev phase, the oscillations are absent. This quantity is expected to be constant and equal to +1 in the thermodynamic limit, but this is not observed because of the finite size. Adding the small disordered field, we further probe the

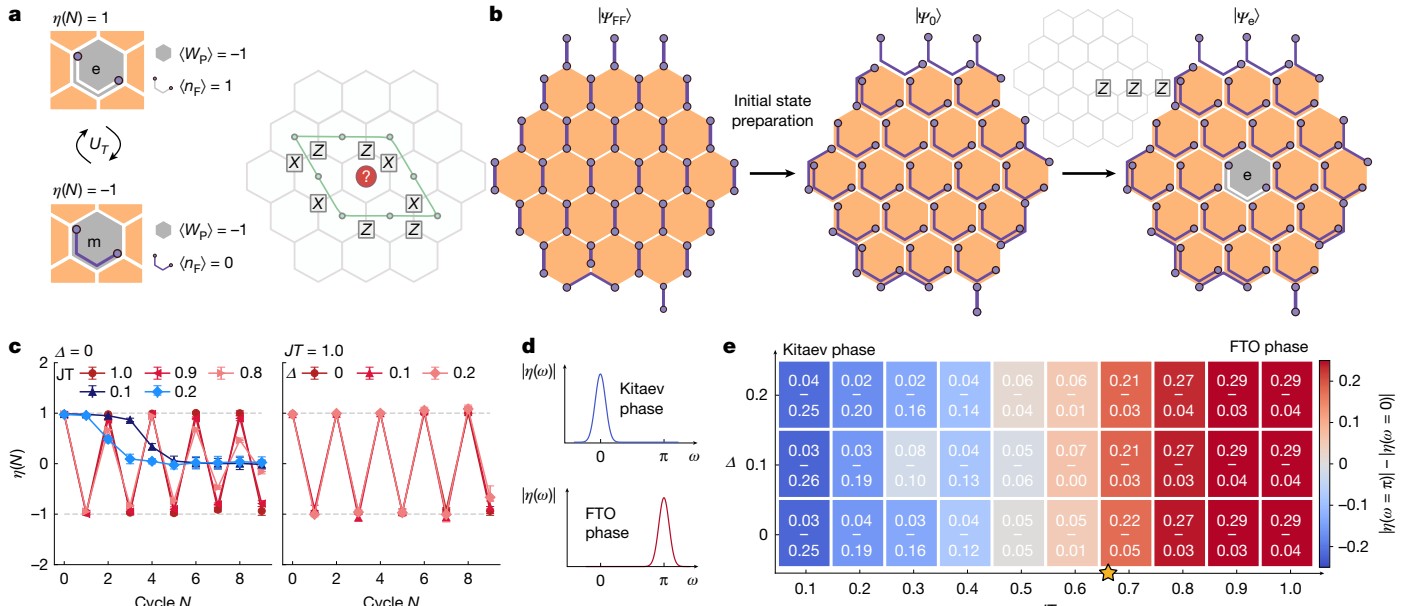

**Fig. 4 | Bulk non-equilibrium topological order. a**, Sketch of $e$ and $m$ anyons, which are transmuted in the bulk under the Floquet driving. Measuring the electric loop operator gives a positive or negative sign for the $e$ and $m$ anyons, respectively. **b**, Initial state preparation. Starting from the flux-free state $|\Psi_{FF}\rangle$, we rearrange the fermions to prepare the state $|\Psi_0\rangle$. Applying three $Z$ operators creates an $e$ anyon in the centre of the system, yielding $|\Psi_e\rangle$. The paired anyon is pushed out of the system on the right-hand side. **c**, Bulk invariant showing oscillations in the FTO phase (red), which are absent in the Kitaev phase (blue). Oscillations in the invariant are robust to integrability-breaking disorder on the time scale of the experiments. Error bars correspond to $2\sigma$ obtained by jackknife resampling (for details, see Methods). **d**, In the Kitaev phase, the Fourier transform of the order parameter $\eta(\omega)$ is expected to peak at zero frequency, whereas it peaks at $\pi$ in the FTO phase. **e**, Phase diagram as a function of the detuning $JT$ and disorder $\Delta$ obtained from the difference of the Fourier components at $\pi$ (top number) and 0 (bottom number). The yellow star marks the critical point of the exactly solvable model ($\Delta = 0$). Data taken with dynamical decoupling and randomized compiling for CZ gates ($N_{twirling} = 20$, $N_{shots} = 10^6$ and $N_{disorder} = 20$, 58 qubits).

robustness of the FTO phase in Fig. 4c. Additional datasets for higher disorder at $JT = 1$ are presented in the Methods. Our data are robust to increasing disorder strength. In the FTO phase, the Fourier transform of the order parameter peaks at $\pi$, whereas in the Kitaev phase at zero, see sketch Fig. 4d. The difference $|\eta(\omega = \pi)| - |\eta(\omega = 0)|$ is thus positive in the FTO phase and negative in the Kitaev phase. Tuning $JT$ and $\Delta$, we map out a pre-thermal non-equilibrium phase diagram in Fig. 4e.

## Outlook

The landscape of non-equilibrium phases of matter is largely unexplored so far. Digital quantum processors provide an ideal platform to reveal the highly entangled dynamical phases therein. The non-equilibrium topological order that we have probed in this work encapsulates a fundamentally unique phenomenology that is forbidden in thermal equilibrium. While many equilibrium phases are known and theoretically understood, the prospects for non-equilibrium phases of matter are still largely open. New order parameters, possibly akin to our non-equilibrium loop order parameter, need to be developed to characterize these dynamical phases of matter. Our interferometric probes for imaging the dynamical transmutation of anyons provide a new approach towards exploring highly entangled non-equilibrium phases of matter.

While completing this project, we became aware of related research that investigates the equilibrium Kitaev phase and its associated emergent fermionic dynamics, implemented on a neutral-atom quantum computing platform[50].

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

## Methods

### Floquet Kitaev model

The Floquet Kitaev model is well understood because of a Majorana representation of the spins, which shows the free-fermion solvability of the model[6,39]. On each site, four Majorana operators $\{c, b^x, b^y, b^z\}$ can be defined, such that the Pauli operations on the spin sites are

$$X_j = \mathrm{i}c_j b_j^x, \quad Y_j = \mathrm{i}c_j b_j^y, \quad Z_j = \mathrm{i}c_j b_j^z, \tag{9}$$

where $c_j^2 = 1$ and $\{c_j, c_k\} = 0$ for $j \neq k$, and similarly for $b_j^\alpha$ (Extended Data Fig. 1a). This representation enlarges the Hilbert space, and the physical Hilbert space corresponds to the set of states $|\Psi\rangle$ such that $c_j b_j^x b_j^y b_j^z |\psi\rangle = |\psi\rangle$ for all $j$. The model has a set of conserved quantities $u_{jk} = \mathrm{i}b_j^{\alpha_{jk}} b_k^{\alpha_{jk}}$, where $\alpha_{jk}$ is the Pauli operator associated with that bond. Note that these are conserved not just stroboscopically, but at all points throughout the drive. These can be thought of as a $\mathbb{Z}_2$ gauge field, and products of these operators around closed loops correspond to physical gauge-invariant conserved quantities, the fundamental instances of which are the flux operators shown in Extended Data Fig. 1a:

$$W_P = \prod_{\langle i,j \rangle \in \bigcirc} u_{ij}. \tag{10}$$

These measure the flux through each of the plaquettes, and we refer to $W_P = +1$ as flux-free. Each of the driving terms is then of the form

$$\exp\left\{-JT\frac{\pi}{4} \sum_{\langle ij \rangle} u_{ij} c_i c_j\right\}, \tag{11}$$

which is quadratic in the $c_i$-Majorana fermion operators. These correspond to hopping operators for the $c$-fermions and perform a Majorana swap when $JT = 1$ (M-SWAP). This driving can be efficiently implemented on a quantum processor using C-PHASE gates and single-qubit rotations, as shown for the example of a single hexagonal plaquette in Extended Data Fig. 1b.

To monitor the dynamics of the emergent Majorana modes, we define measurable operators by pairing them with complex fermions. The paired fermion operators are then defined as $f_{jk} = (c_j + \mathrm{i}\phi_{jk} c_k)/2$, where

$$\phi_{jk} = \prod_{\langle l,m \rangle \in \mathrm{path}\,(j,k)} u_{lm} \tag{12}$$

is a gauge string consisting of a product of the gauge fields along a path connecting the paired sites. This leads to a density operator

$$n_F(j,k) = \frac{1}{2}(\mathrm{i}\phi_{jk} c_j c_k + 1), \tag{13}$$

which can be written as a Pauli string. Under the Floquet dynamics, these paired fermions move around the system, as shown in Extended Data Fig. 1c.

### Experimental procedures

CZ and C-PHASE gates are implemented by setting the qubit detuning close to the anharmonicity and harnessing a diabatic $|11\rangle \rightleftarrows |20\rangle$ swap to generate an arbitrary C-PHASE angle with minimal leakage[51]. Dominant errors come from CZ/C-PHASE entangling gates and final readout[52] (Extended Data Fig. 2a). The Snake optimizer[53,54] has been used to optimize qubits, coupler and readout parameters. A smaller contribution to the total error comes from the single-qubit microwave gates, which are calibrated using the Optimus calibration tools of Google[55,56].

### Dynamical decoupling

Idle qubits are exposed to errors over time. In particular, this is important for Hadamard tests such as the Floquet braiding experiment. In contrast to all other qubits, the ancilla is not part of the Floquet time evolution and is idle during this time. To quantify those errors under experimental conditions, we use the same circuit as in the actual Floquet braiding experiment but do not couple the ancilla to the system (Extended Data Fig. 2b). Therefore, the ancilla should remain in the $|+\rangle$ state as no other operation is performed on it. In the Hadamard test, we would measure the expectation value of $\langle X_A \rangle$ and $\langle Y_A \rangle$ and we, therefore, probe those two expectation values here. In Extended Data Fig. 2c, we would expect to measure $+1$ at all times for $\langle X_A \rangle$ and $0$ for $\langle Y_A \rangle$. However, we observe that $\langle X_A \rangle$ decreases quickly, whereas $\langle Y_A \rangle$ increases at the same time. To mitigate this effect, we apply a simple form of dynamical decoupling[57] by randomly applying $X$-$X$ or $Y$-$Y$ sequences on the idle ancilla. Extended Data Fig. 2d shows that this technique effectively reduces these errors rather well.

### Randomized compiling

To address CZ-gate dependent errors, which can arise from imperfect gate calibration, we have applied randomized compiling to all CZ gates in all experiments. This technique[58,59] involves inserting sets of Pauli operators before and after the CZ gates, leaving the total circuit invariant while significantly enhancing the overall performance. We repeated all experiments for 20 different twirling sets.

### Noisy simulation

To further explore the effects of noise on our results, we perform noisy quantum trajectory simulations and compare them with experimental data (Extended Data Fig. 2e) of a two-plaquette setup, which is shown on the left-hand side. Because we use randomized compiling, we approximate CZ gate errors as a two-qubit depolarizing channel. In the simulation, we apply different error rates to each pair of qubits, obtained from two-qubit XEB experiments conducted just before the experimental data in Extended Data Fig. 2e was collected. We also include amplitude damping on the ancilla during the Floquet cycles to account for $T_1$ decay, assuming a $T_1$ of 25 µs. Finally, we include readout errors of 0.4% on the $|0\rangle$ state and 2.5% on the $|1\rangle$ state. This error model produces results that are qualitatively consistent with the experimental data: the overlap shows a decaying magnitude, $r$. Together with the results in Extended Data Fig. 2b–d, this agreement suggests the described error model with an additional coherent phase drift on the ancilla qubit (strongly mitigated by dynamical decoupling) reproduces our experimental observations.

### Extended data and derivations

**Preparation of initial states.** To prepare a flux-free state, we use the circuit shown in Extended Data Fig. 3a for the seven-plaquette setup and the circuit shown in Extended Data Fig. 4 for the large system, which was introduced in refs. 16,42, where $H$ is the Hadamard gate

$$H = \frac{1}{\sqrt{2}}\begin{bmatrix} 1 & 1 \\ 1 & -1 \end{bmatrix}$$

and $S$ denotes the phase gate given by

$$S = \begin{bmatrix} 1 & 0 \\ 0 & i \end{bmatrix}.$$

After this initial preparation, fermions are paired vertically on $z$-bonds. In the main text, we denote this flux-free state as $|\psi_{FF}\rangle$. For all experiments, we first prepare this flux-free state and then rearrange the fermions depending on the initial state needed. To preserve the fluxes, fermions can be rearranged by applying $\mathrm{e}^{-\mathrm{i}JT\frac{\pi}{4}(\alpha_{jk})_j(\alpha_{jk})_k}$ with $JT = 1$ so that those operators correspond to swap operations of $c$-Majoranas of the respective bond (M-SWAP). Choosing the operator depending on the type of bond ($x$, $y$ or $z$) preserves the flux. Extended Data Fig. 3b shows the step-by-step rearrangement of fermions for the initial state of Fig. 1b.

The circuit for the flux-free state preparation of the 58-qubit setup used in the *e-m* transmutation experiment, as well as the two layers of M-SWAP gates used for rearranging the fermions, is shown in Extended Data Fig. 4. For all other experiments, state preparation is detailed in the figures in the main text. Generally, we have chosen initial states to ensure that the circuits are as shallow as possible while yielding the clearest experimental results. For example, in the case of the spectral function, the only requirement for the initial state, apart from being flux-free, is that the protruding bond hosts a perfectly occupied fermion. Beyond that, the specific pairing of Majoranas in the rest of the system is irrelevant. As such, starting directly from the flux-free state yields the shallowest circuit. Notably, we have this freedom in the initial state, as we are probing properties of the Floquet operator rather than fine-tuned initial states.

**Data analysis.** For all datasets, we apply jackknife resampling before further analysis. The raw data for each experiment consists of 20 twirling sets, and each twirling set contains many measured data points. To estimate statistical uncertainties, we construct jackknife samples by systematically leaving out one entire twirling set at a time. More precisely, let $\theta_i$ denote the average for the $i$th twirling set. Then the $i$th jackknife sample $j_i$ is given by

$$j_i = \frac{1}{N_{\text{twirling}} - 1} \sum_{k \neq i}^{N_{\text{twirling}}} \theta_k. \tag{14}$$

This procedure yields $N_{\text{twirling}} = 20$ jackknife samples $j_i$. When analysing a quantity $f$ that depends on multiple measured variables, we construct jackknife samples for each variable on its own. Suppose our desired analysis function is a multivariate function $f$, then for each jackknife index $i$, we have a set of jackknife samples $(j_i, l_i, \ldots)$ corresponding to all variables. Applying $f$ to these jackknife samples yields 20 samples:

$$F_i = f(j_i, l_i, \ldots). \tag{15}$$

Finally, the jackknife mean and standard error are given by

$$\bar{F} = \frac{1}{N_{\text{twirling}}} \sum_{i=1}^{N_{\text{twirling}}} F_i \tag{16}$$

$$\sigma = \sqrt{\frac{N_{\text{twirling}} - 1}{N_{\text{twirling}}} \sum_{i=1}^{N_{\text{twirling}}} (F_i - \bar{F})^2}. \tag{17}$$

In all figures, error bars denote $\pm 2\sigma$ as a measure of spread. As a concrete example, in the braiding experiment, we measure the real and imaginary parts of the complex amplitude separately. Jackknife resampling is applied to each dataset independently, yielding 20 jackknife estimates for the real part and 20 for the imaginary part. The corresponding real and imaginary sets are then paired up for each jackknife index to reconstruct the full complex amplitude. From these pairs, we compute the radial and angular components, and finally estimate the mean and standard error for each observable across this jackknife ensemble.

**Floquet braiding.** In Fig. 2, we present post-selected data of the Floquet braiding experiment. The Floquet drive without disorder conserves the fluxes in the system, allowing us to use those operators for post-selection. As all flux operators commute with each other, they can, in theory, be measured simultaneously. However, this requires a deep two-site gate circuit in general. Therefore, it is important to balance between adding extra layers and post-selecting on as many plaquettes as possible.

For two plaquettes, Extended Data Fig. 5a shows how a single additional layer of CZ gates allows us to simultaneously measure the fluxes of two adjacent plaquettes. For example, to simultaneously measure the fluxes of two plaquettes sharing a $y$-bond, a CZ gate is first applied to the shared bond, rotating the two sites into the joint eigenbasis of the corresponding operators. By measuring the operators shown on the right-hand side, we can then measure the two fluxes at the same time. To simultaneously extract the six flux operators shaded in orange in Extended Data Fig. 5b, we apply this circuit on all shared bonds. Depending on the bond, additional rotations are required to align with the eigenbasis of $XY$ and $YX$ ($z$-bond) or $ZY$ and $YZ$ ($x$-bond).

In Extended Data Fig. 5c, we present both post-selected and non-post-selected data, demonstrating that post-selection significantly improves the radial component.

**Spectral function.** The Floquet Kitaev model in the FTO phase hosts a chiral edge mode of Majorana fermions. To investigate the stability of these edge modes, we measure a Majorana spectral function along the edge of the system. Experimentally, this measurement is achievable by evaluating Pauli strings along the edge.

The key idea is to add an extra site to the system, which is not subject to time evolution. By initializing a fermion on this protruding bond, only the end of the fermion within the driven part of the system will move along the edge, whereas the other end remains stationary outside the system. This setup allows us to compute the expectation value

$$\langle \Psi_0 | P_j(N) P_0(0) | \Psi_0 \rangle. \tag{18}$$

This expectation value relates to a Majorana spectral function of the form

$$\langle \Psi_0 | c_j(N) c_0(0) | \Psi_0 \rangle. \tag{19}$$

First, we note that the two Pauli strings overlap on the protruding bond. By rewriting these Pauli strings in terms of Majorana fermions, as shown in Extended Data Fig. 6a, we observe that the non-time-evolved parts cancel out to one. This results in a Majorana–Majorana spectral function, allowing us to probe the stability of the chiral edge modes.

In Extended Data Fig. 6b, we present exact diagonalization data for the spectral function. In contrast to the experimental data presented in Fig. 3d, the signal does not decay for $JT = 1$. We attribute the decay in the experiment to a combination of coherent errors and decoherence in the device. Still, the feature of a chiral Majorana edge mode is well observed experimentally. In Extended Data Fig. 6d, we present experimental data without post-selection. By contrast, in Fig. 3d, we present post-selected data, in which we have used the central plaquette of the setup for post-selection. Note that it is not possible to post-select on more plaquettes without a large overhead of two-site gates. Extended Data Fig. 6c shows the standard error of the data shown in the main text. To obtain these uncertainties, jackknife resampling was performed over the twirling sets before computing the spectral function. We have applied a cosine window function to the sixth power to the data before Fourier transforming in time. At the fine-tuned point of $JT = 1$, Majoranas are perfectly transferred to a neighbouring site. $P_0$ is perfectly transferred into $P_j(4)$ after four time steps. We would thus expect the correlation in equation (18) to be 1. We define the Fourier transform in terms of the translation matrix and its eigenvectors $\hat{F}|v_q\rangle = e^{iq}|v_q\rangle$ at $JT = 1$. The Fourier transform of a state $|\Psi\rangle$ is then given by the overlap

$$|\Psi_q\rangle = \langle v_q | \Psi \rangle, \tag{20}$$

where $N$ is the number of edge sites.

**e-m Transmutation.** In the bulk, the FTO phase is characterized by $\mathbb{Z}_2$ topological order, hosting $e$, $m$ and $\psi$ anyons[6]. This property can directly be observed at $JT = 1$ for fermions paired along the diagonal of a plaquette, depicted as state 2 in Extended Data Fig. 1c. In this case,

this bulk fermion evolves back to its original position after driving twice. In the following, we will use the convention that a bulk fermion is occupied if the Pauli string which measures the occupation $n_F$, defined as in state 2, is equal to +1. The fermion parity $P_F = (-1)^{n_F} = \pm 1$ picks up the flux of the respective plaquette under driving

$$U_T^\dagger P_F U_T = W_P P_F. \tag{21}$$

If the flux through a plaquette is +1, the fermion occupation in that plaquette remains constant during time evolution. However, if the flux is $W_P = -1$, the fermion occupation will alternate after each driving cycle, resulting in a periodicity of $N = 2$. In general, a $\psi$ anyon corresponds to an occupied bulk fermion $n_F = 1$ and a pair of them can be created on top of the flux-free state by applying a Pauli string, as explained in Extended Data Fig. 7a. Conversely, creating a flux defect $W_P = -1$ without changing the fermion occupation $n_F = 0$ corresponds to an $m$ anyon. An example of the creation of two $m$ anyons is shown in Extended Data Fig. 7b. The combination of a flipped flux $W_P = -1$ and an occupied fermion $n_F = 1$ leads to an $e$ anyon (see, for example, Extended Data Fig. 7c). In the FTO phase, an $e$ anyon in the bulk will evolve into an $m$ anyon and after $N = 2$ cycles back into an $e$ anyon. To measure this $\mathbb{Z}_2$ topological order, we can measure an $e$ loop around the central anyon. This $e$ loop is an $e$ anyon dragged around, for example, a plaquette and then annihilated in the end. This operator commutes with an $e$ anyon, thus giving +1, but anticommutes with the $m$ anyon, thus giving −1.

In the main text, we present results of the bulk invariant (Fig. 4c), which shows clear oscillations in the FTO phase that are absent in the non-Abelian Kitaev phase.

In Extended Data Fig. 8a, we present the time evolution of the $e$ loop operator for the excited and non-excited states separately. As the expectation values tend to approach zero and only a finite number of measurements are available, we introduce a small shift of 0.01 in the denominator of the order parameter $\eta(N)$ to ensure numerical stability. We first apply jackknife resampling to the twirling sets and then compute the mean of the ratio by averaging over those sets:

$$\eta(N) = \frac{1}{N_{\text{twirling}}} \sum_{i=1}^{N_{\text{twirling}}} \frac{\overline{\langle O(N) \rangle_{e_i}}}{|\overline{\langle O(N) \rangle_0}|_i + 0.01}, \tag{22}$$

where the numerator and denominator are computed from the same jackknife subset $i$, corresponding to measurements with and without anyons, respectively. The standard error of $\eta(N)$ is extracted from the distribution of jackknife samples $\eta(N)_i$ using equation (17). Extended Data Fig. 8b shows the order parameter $\eta(N)$ for different combinations of $JT$ and $\Delta$. Applying a cosine window function to the fourth power and Fourier transforming those datasets yields the spectra shown in Extended Data Fig. 8c. In the FTO phase, we expect a peak at π, whereas in the Kitaev phase at zero. Thus, the difference of $|\eta(\omega = \pi)| - |\eta(\omega = 0)|$ has different signs in the two phases. This difference is shown in Fig. 4d. In Extended Data Fig. 8d, we give standard errors in addition to the mean values at zero and π momentum.

To investigate the impact of disorder, we performed measurements at $JT = 1$ with increasing disorder strength $\Delta$ (Extended Data Fig. 9). For small disorder ($\Delta \lesssim 2.4$), the order parameter shows oscillatory behaviour, indicative of the FTO phase, although the amplitude gradually decreases as $\Delta$ increases. At higher disorder strengths, these oscillations become suppressed and eventually disappear. The observed behaviour suggests a crossover or transition towards a localized phase at strong disorder. However, owing to limited experimental resources, we averaged only over 50 disorder realizations for each data point. As a result, we cannot make definitive claims about the nature of the transition.

## Matrix-product state simulations

In the following, we simulate the $e$-$m$ transmutation using matrix-product states (MPS) to get an estimate for the computational hardness of simulating the experiments presented in the main text. We use the Python library TeNPy for our simulations[60].

For our simulations, we consider the experiment in Fig. 4. To order the sites into a one-dimensional geometry suitable for MPS simulations, we start from the site at the bottom of the bottom-left hexagon and move up diagonally along $y$- and $z$-bonds until we reach the boundary of the system, before moving to the next diagonal and starting at the bottom. This way, sites sharing a $y$- or $z$-bond remain neighbours in the 1D geometry, and only $x$-bonds become long-range. The initial state $|\psi_0\rangle$ (Fig. 4b, middle) is a stabilizer state, so we use DMRG[60,61] to obtain the simultaneous eigenstate of all its stabilizer Pauli strings without having to run the full state preparation circuit. The $e$ anyon can be created, as in the experiment, by applying onsite $Z$-gates to the MPS tensors. The gates on the $x$-bonds can be written as a matrix-product operator (MPO) with bond dimension two acting between the two (non-neighbouring) sites. Applying the MPO to the MPS doubles the bond dimension, after which we truncate back to the maximal bond dimension using SVDs. As all sites sharing a $y$- or $z$-bond are neighbours, we can apply these gates as in TEBD[62], sweeping left to right when applying the gates on the $y$-bonds and sweeping back right to left when applying the gates on the $z$-bonds. If we include the disordered $Z$-field, the fourth time step consists only of single-qubit rotations that can be directly applied to the corresponding MPS tensor.

First, we consider the fine-tuned point with $JT = 1.0$ and without disorder (that is, $\Delta = 0$). Extended Data Fig. 10a shows the maximal entanglement entropy of all bonds at each time step, for the initial state with an $e$ anyon. The entanglement entropy for the state without an $e$ anyon is the same, so we only show the data for the initial state with an $e$ anyon. Different marker shapes and shades of red denote the different bond dimensions used for the simulation. The entanglement entropy saturates at $S = 9 \log(2)$, indicating that at least a bond dimension of $\chi = 512$ is needed. The simulation data with $\chi = 512$ and $\chi = 1{,}024$ do not overlap because intermediate steps in the Floquet cycle can further increase the bond dimension, which then gets truncated in the simulation with $\chi = 512$. As the initial state is a stabilizer state, we can calculate the entanglement entropy (and the required bond dimension) for larger system sizes. Extended Data Fig. 10a (inset) shows the entanglement entropy of the initial state against the total number of qubits when cutting the system from the top-left plaquette to the bottom-right plaquette for an increasing number of rings. Asymptotically, the entanglement entropy scales as $S(N_Q) = \sqrt{2/3 N_Q} \log(2) + O(1)$ (dashed line), where $N_Q$ is the total number of qubits in the system. This translates to a required bond dimension of $\chi = \Theta\left(2^{\sqrt{2/3 N_Q}}\right)$, which scales superpolynomially with the number of qubits. Extended Data Fig. 10b shows, from left to right, the expectation value of the loop operator when starting from the state with an $e$ anyon, when starting from the state without an $e$ anyon, and the order parameter $\eta(N)$ defined in equation (8), which is the ratio of the two expectation values. Using a bond dimension $\chi = 1{,}024$ or larger, we see that the expectation value of the loop operator persistently oscillates between +1 and −1 if we start from the state with an $e$ anyon, and remains perfectly at +1 if we start without an $e$ anyon. Any smaller bond dimension leads to a decay of the expectation values, which, remarkably though, almost perfectly cancels when taking their ratio such that the order parameter does not deviate strongly for smaller bond dimensions.

Next, we consider the case where $JT = 1.0$ and we include the disordered field with strength $\Delta = 0.2$. Note that compared to the previous fine-tuned example with $JT = 1.0$ and $\Delta = 0$, the time evolution now neither corresponds to a Clifford circuit, which could be efficiently simulated using stabilizer methods, nor does it map to a time evolution of non-interacting fermions, which could also be simulated efficiently; thus, we need a simulation method such as MPS that can deal with interacting systems in general. Extended Data Fig. 10c shows the maximal entanglement entropy of all bonds at each time step, averaged over 20 disorder realizations, the error bars show the standard deviation.

The different marker shapes and shades of red again denote the different bond dimensions used for the simulation. The plot of the entanglement entropy looks similar to before; only because now the state is no longer a stabilizer state, the entanglement entropy is no longer a perfect multiple of $\log(2)$ and grows above $S = 9\log(2)$ for the simulations with $\chi = 1{,}024$ and $\chi = 2{,}048$. Extended Data Fig. 10d shows the expectation value of the loop operator when starting from the state with an $e$ anyon, when starting from the state without an $e$ anyon, and their ratio, which is the order parameter $\eta(N)$ defined in equation (8). As we are away from the fine-tuned point, we now see the expectation value of the loop operator decay in time both when starting from a state with an $e$ anyon and without. For short times, this decay cancels approximately when taking their ratio, showing the oscillations of the order parameter between $+1$ and $-1$. At later times, the two loop expectation values become small and the MPS simulation becomes unreliable because of truncation errors, so the order parameter becomes unstable. The smaller bond dimensions show deviations from the $\chi = 2{,}048$ simulation, indicating that the simulation is not yet converged. The $\chi = 1{,}024$ simulations still show deviations from $\chi = 2{,}048$ as early as four cycles in, indicating the need for a large bond dimension for the simulation. Note that the fact that the absolute value of the order parameter becomes larger than 1 at late times is not necessarily due to truncations in the MPS simulations but can be an artefact of the finite system. Only in the thermodynamic limit do we expect the order parameter to perfectly oscillate between $+1$ and $-1$ away from the fine-tuned point.

In general, we see that the entanglement entropy of the state grows linearly in time, initially with a growth rate of $\log(2)$ and followed by a slower rate for the evolution with disorder. Although much of the entanglement entropy coming from our choice of initial state could, in principle, be captured by two-dimensional tensor networks, the subsequent growth of the entanglement entropy is generic and expected to eventually violate a two-dimensional area law[63–66]. Thus, we anticipate that classical simulation of this setup will become intractable for any tensor-network-based approach for large systems and late times. Even for the system sizes accessible to us on the quantum processor, we cannot achieve convergence even with the largest bond dimension $\chi = 2{,}048$ that we have simulated. Furthermore, as at every Floquet cycle the circuit now introduces both non-Clifford operations and interacting fermionic gates, we expect the simulation to also be hard with techniques building on the Clifford or free-fermion structure. Note, however, that we have not explicitly checked these or other methods (for example, projected entangled-pair states, isometric tensor-network states and neural network states) directly.

## Data availability

The data presented in this study are available at Zenodo[67] (https://doi.org/10.5281/zenodo.15837492).

## Code availability

The codes used in this study are available at Zenodo[67] (https://doi.org/10.5281/zenodo.15837492).

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

**Acknowledgements** M.W. thanks A. Vishwanath and E. Berg for their discussions. A.G-S. acknowledges support from the Royal Commission for the Exhibition of 1851 and support from the UK Research and Innovation (UKRI) under the Horizon Europe funding guarantee (grant no. EP/Y036069/1) of the UK government. M.W., B.J., F.P. and M.K. acknowledge support from the Deutsche Forschungsgemeinschaft (DFG, German Research Foundation) under the Excellence Strategy of Germany (EXC–2111–390814868, TRR 360-492547816 and DFG grant nos. KN1254/1-2 and KN1254/2-1), the European Research Council (ERC) under the Horizon 2020 research and innovation programme of the European Union (grant agreement nos. 851161 and 771537), as well as the Munich Quantum Valley, which is supported by the Bavarian state government with funds from the Hightech Agenda Bayern Plus. M.W. and F.P. acknowledge support from the DFG Research Unit FOR 5522 (project id 499180199). All experiments were conducted remotely on 72-qubit Google Sycamore (Figs. 1–3) and Willow (Fig. 4) processors[68,69], with access provided by the Google Cloud Quantum Engine. Calibration and support were provided by the Quantum Hardware Residency Program. M.W. thanks S. Kumar and R. Oliver for calibrations during the review process. We thank the Google Quantum AI team for providing the quantum systems and support that enabled these results. The views expressed in this work are solely those of the authors and do not reflect the policy of Google or the Google Quantum AI team.

**Author contributions** P.R., M.K., A.G.-S and F.P. conceived the project. Experimental data collection protocols were conceived and implemented by M.W. with assistance from A.G.-S., T.A.C. and E.R. Device calibration was done by T.A.C., E.R. and N.M.E. Theoretical models were developed by M.W., T.A.C., B.J., A.G.-S., M.K. and F.P. Numerical simulations were performed by M.W. and B.J. All authors wrote the paper.

**Funding** Open access funding provided by Technische Universität München.

**Competing interests** The authors declare no competing interests.

**Additional information**
**Correspondence and requests for materials** should be addressed to P. Roushan, M. Knap, A. Gammon-Smith or F. Pollmann.

## a Majorana representation

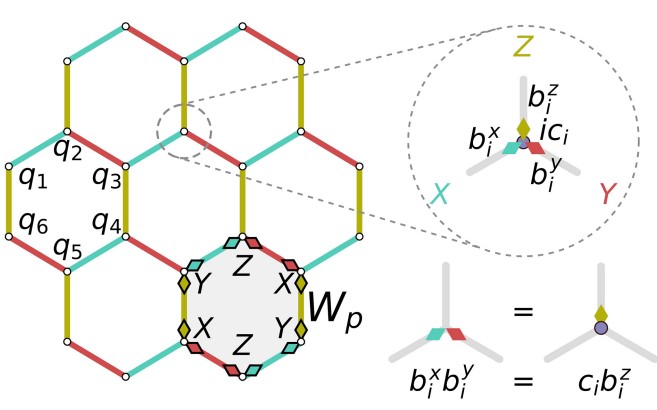

$$U_T = e^{-iJT\frac{\pi}{4}\sum Z_iZ_j}e^{-iJT\frac{\pi}{4}\sum Y_iY_j}e^{-iJT\frac{\pi}{4}\sum X_iX_j}$$

## b Quantum circuit

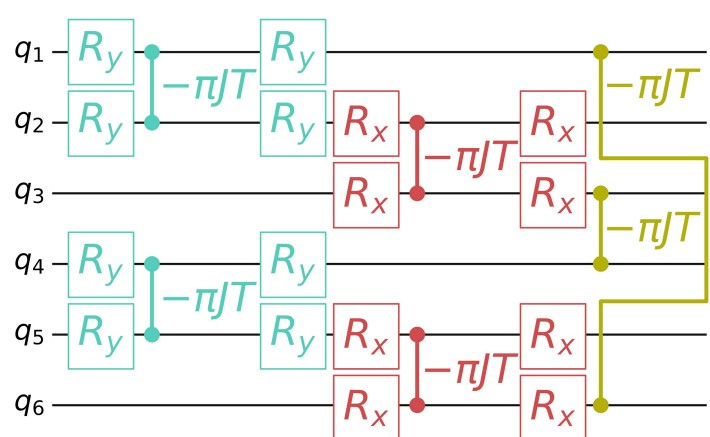

## c Fermion dynamics ($JT = 1.0$)

state 1

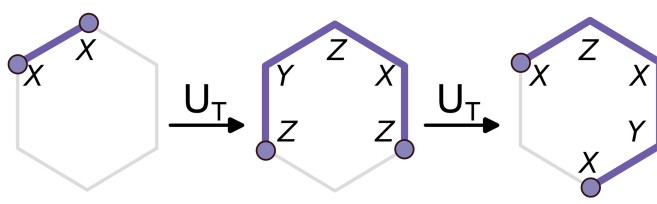

state 2

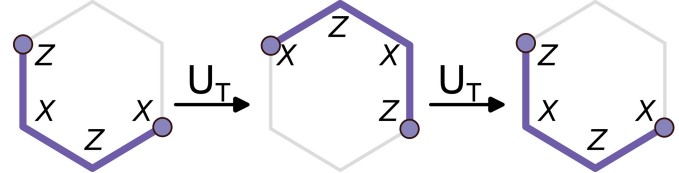

**Extended Data Fig. 1 | Overview. a**, Majorana representation. **b**, Quantum circuit for one plaquette for one Floquet cycle in terms of single qubit rotations and C-PHASE gates. **c**, Two examples of fermion dynamics at the fine-tuned point of $JT = 1$. State 2 is invariant under two driving cycles.

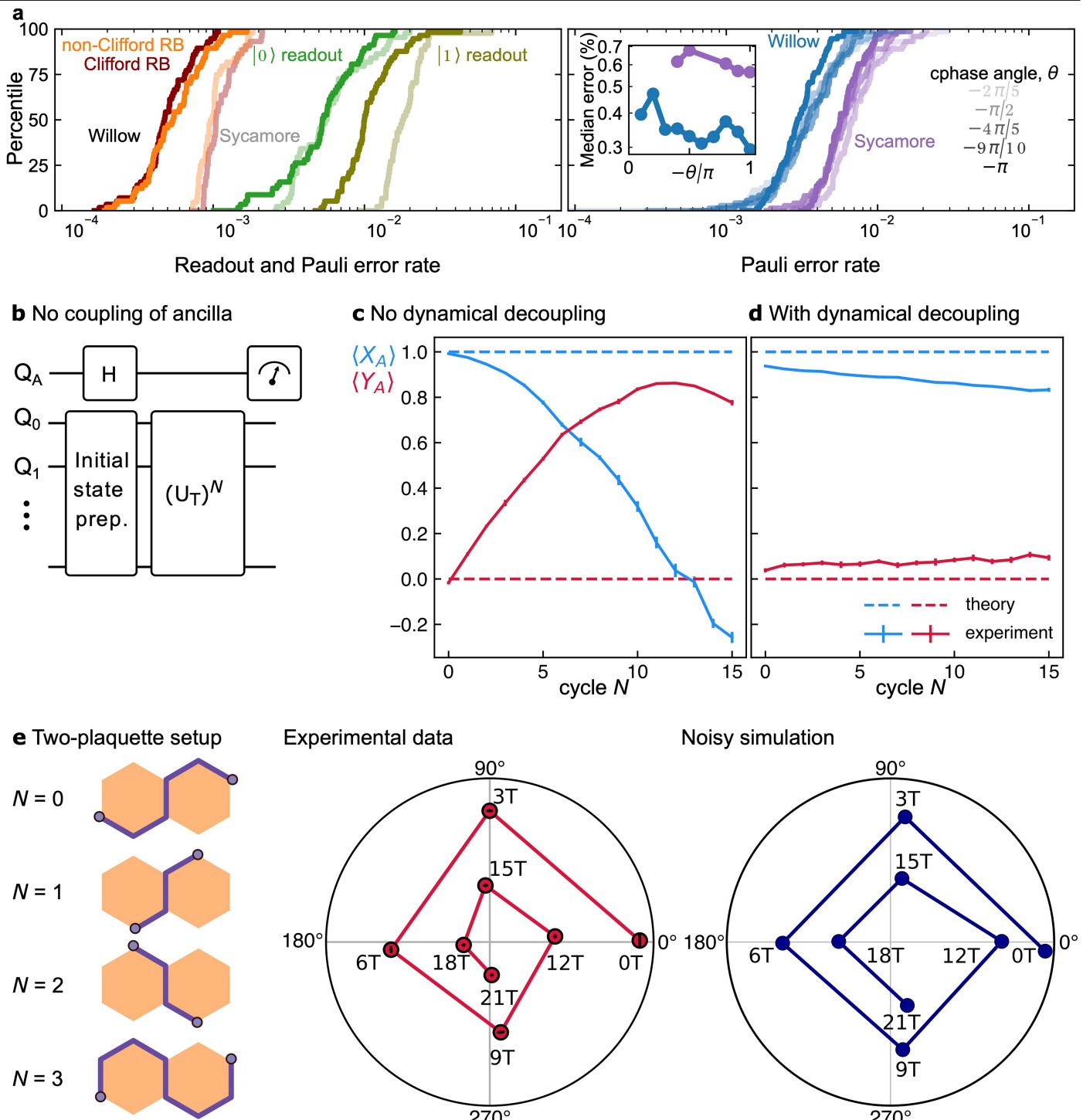

**Extended Data Fig. 2** | See next page for caption.

**Extended Data Fig. 2 | Experimental fidelities, dynamical decoupling and noisy simulation. a, Experimental fidelities.** Representative cumulative distribution functions of relevant gate and measurement errors. Single-qubit Clifford and non-Clifford Pauli errors, determined from randomized benchmarking (RB), are shown in red and orange with median errors of 0.10% and 0.096%, respectively, on the Sycamore device and 0.047% and 0.054% on the Willow device. $|0\rangle$ state and $|1\rangle$ state readout errors, determined from sampling random bitstrings, are shown in green and olive with median errors of 0.55% and 1.8%, respectively on the Sycamore device and 0.53% and 1.0% on the Willow device. Inferred CZ Pauli errors ($\theta = -\pi$), determined from cross-entropy benchmarking, are shown in purple (Sycamore) and blue (Willow) with a median error of 0.56% on the Sycamore device and 0.29% on the Willow device. Distribution functions for inferred errors of CPHASE gates with selected angles used in this work are also shown ($\theta \in \{-9\pi/10, -4\pi/5, -\pi/2, -2\pi/5\}$). Lighter shades correspond to angles closer to zero. The inset shows the median error rates for all CPHASE angles used in this work. To implement non-Clifford RB, we replace the standard depth-$n$ RB sequence, which consists of $n-1$ random Clifford gates and a final Clifford gate that inverts the sequence, with $U_f X U_{\frac{n-1}{2}} \dots X U_0$, where each $U_i$ is a Haar-random single-qubit unitary, and $U_f$ is computed to invert the whole sequence. The error rate thus obtained, which includes approximately equal contributions from the Clifford $X$ gates and the non-Clifford $U_i$ gates, is what is plotted in panel **a** as '1Q non-Clifford'. **b, Dynamical decoupling.** Circuit to test stability of ancilla. To mimic the Floquet braiding experiment, we use the same circuit except for not coupling the ancilla to the system. **c,** Without dynamical decoupling the ancilla is idle during the time evolution and drifts away from being in the $|+\rangle$ state. **d,** Adding dynamical decoupling by randomly applying $X$-$X$ and $Y$-$Y$ pairs to the ancilla resolves those errors. **e, Noisy simulation.** Majorana interferometry as described in Fig. 2c on a two plaquette system is expected to show amplitude oscillations every three cycles (left). Plotting the results as $re^{i\phi}$, the experiments show that $r$ decays with each Floquet cycle, which is reproduced by adding a two-qubit depolarizing noise channel after each CZ gate, amplitude damping on the ancilla to simulate $T_1$ decay, and readout errors (right). Error bars correspond to $2\sigma$ obtained via jackknife resampling. Data taken on Sycamore chip with $N_{\text{shots}} = 5 \times 10^5$, $N_{\text{twirling}} = 20$.

## a Flux-free state preparation

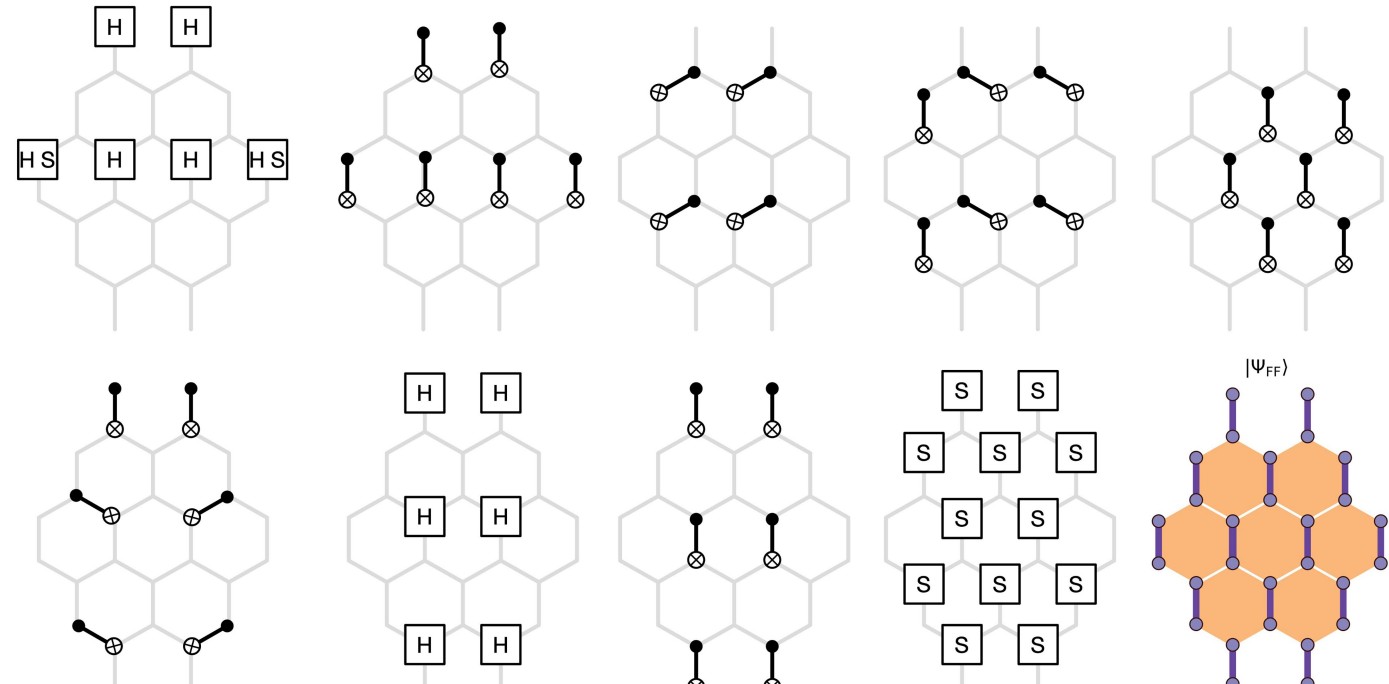

## b State preparation for Fig. 1

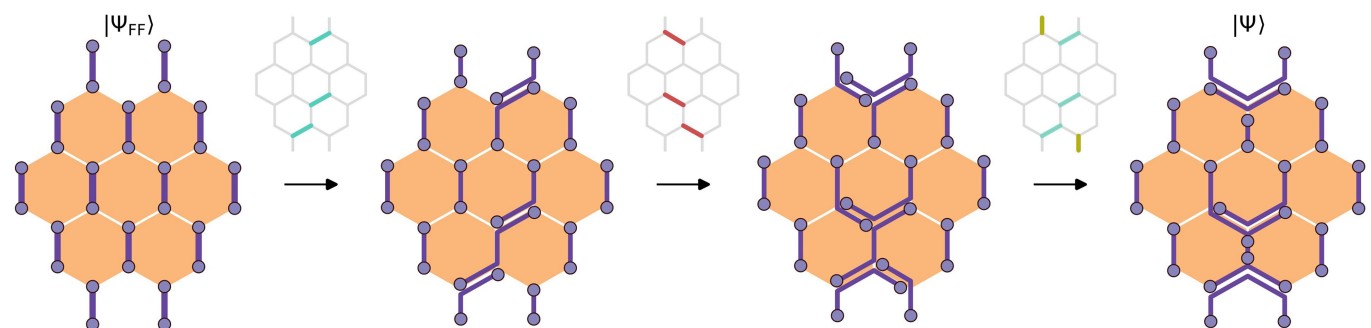

**Extended Data Fig. 3 | State preparation using two-site gates. a**, We have adopted the preparation scheme of refs. 16,42 for the preparation of a flux-free state. In the flux-free state $|\Psi_{FF}\rangle$ all fermions are paired vertically on $z$-bonds, as shown in purple, and all fluxes have an expectation value of one. **b**, Starting from the flux-free state, M-SWAP gates are used to prepare the initial state of Fig. 1b.

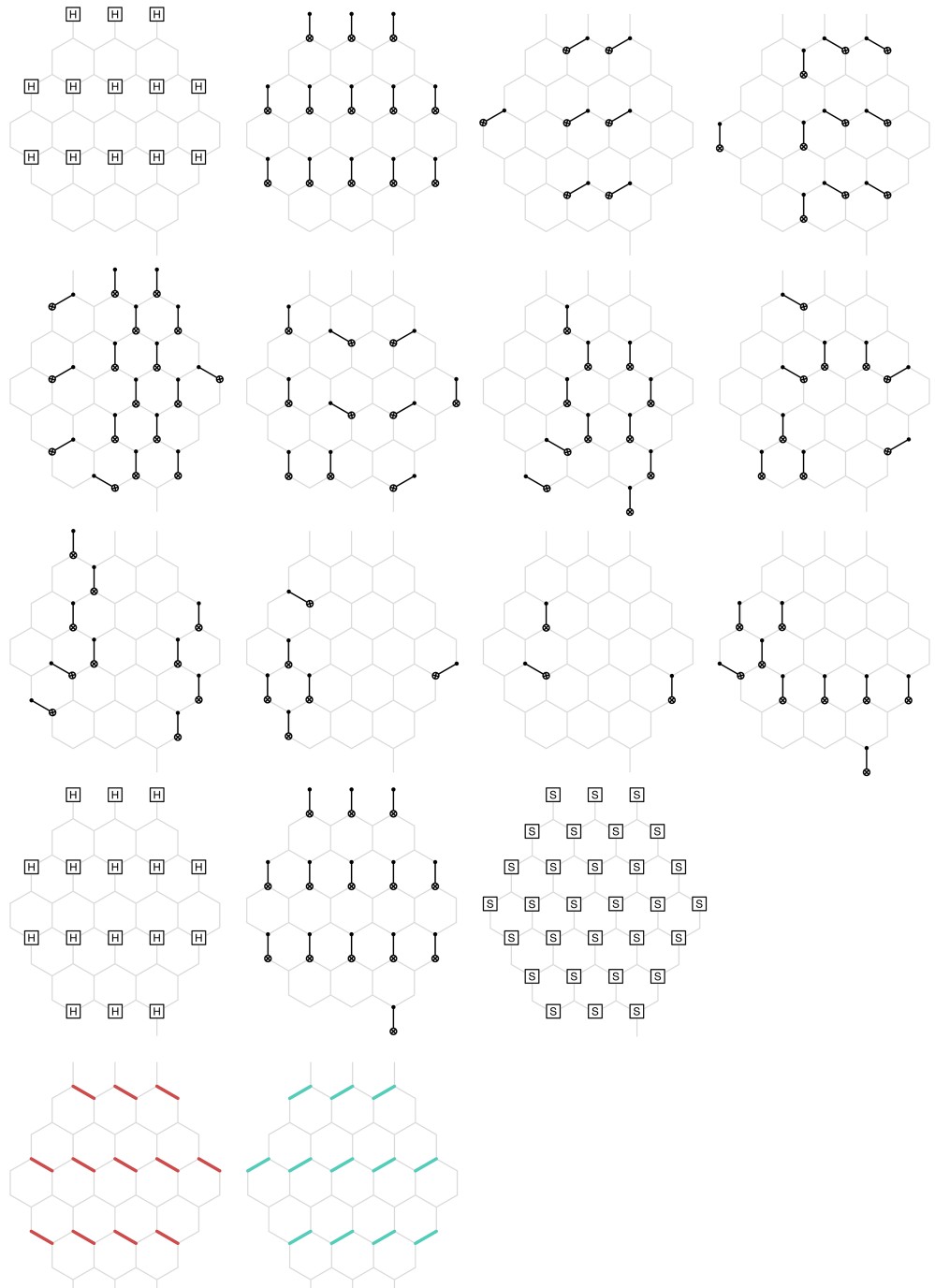

**Extended Data Fig. 4 | Initial state preparation for *e-m* transmutation for Fig. 4.** After preparing the flux-free state, applying M-SWAP gates on *y*- and then *x*-bonds prepares the initial state for the *e-m* transmutation experiment.

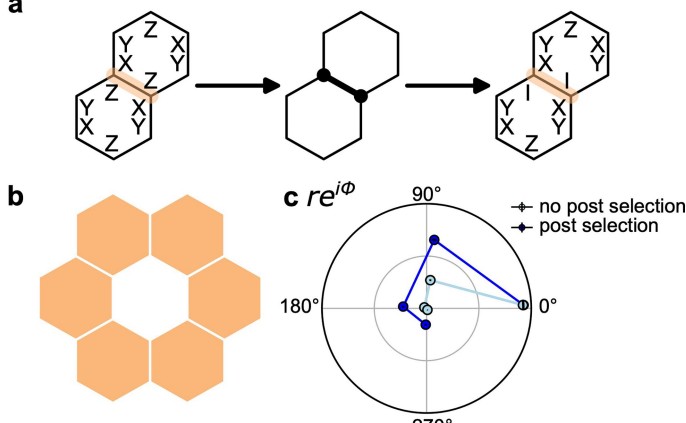

**a**

**b**

**c** $re^{i\Phi}$

— no post selection
— post selection

**Extended Data Fig. 5 | Post selection on fluxes. a**, To measure two flux operators at the same time, one has to rotate the shared bond into an eigenbasis of the operators, e.g., coloured bond. Applying a CZ gate on the shared bond rotates the bond into a basis such that both flux operators can be measured at the same time. In the new basis we have to measure the operators on the right hand side to evaluate the flux. **b**, By applying this scheme to shared bonds, we can simultaneously post-select all six shaded plaquettes using only one additional CZ layer in the Floquet braiding experiment. **c**, Comparison of (non)-post selected experimental data. Error bars correspond to $2\sigma$ obtained via jackknife resampling ($N_{\text{shots}} = 5 \times 10^5$, $N_{\text{twirling}} = 20$).

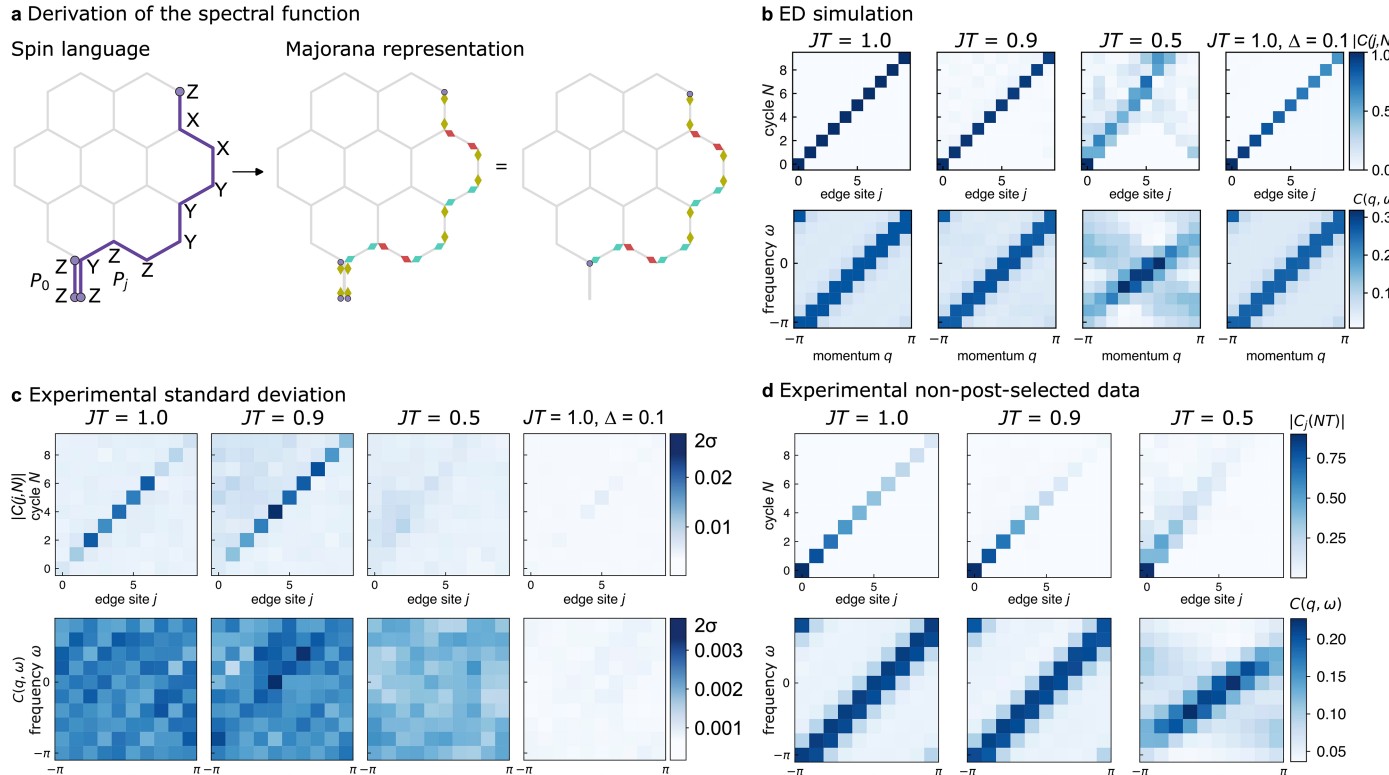

**a** Derivation of the spectral function

Spin language Majorana representation

**b** ED simulation

**c** Experimental standard deviation

**d** Experimental non-post-selected data

**Extended Data Fig. 6 | Details on the spectral function. a**, Example of Pauli strings, which have to be evaluated in order to calculate the spectral function. The purple Pauli string emerges from the black fermion after time evolution of $N = 4$. Pauli strings can be rewritten in terms of Majorana fermions. For the specific example, the decomposition is illustrated. The two fermions overlap on the protruding bond. There, the Majorana operators occur twice and thus multiply to one. Thus one is left with the string of Majorana fermions shown on the right hand side. **b**, Exact diagonalization data. **c**, $2\sigma$ obtained via jackknife resampling of the data shown in the main text Fig. 3e. The spectral function was computed following jackknife resampling over the 20 twirling sets, and standard errors were estimated from the resulting ensemble. **d**, Spectral function without post selection on central plaquette. For the Floquet time evolution without an additional field, the fluxes are conserved and can be used for post selection.

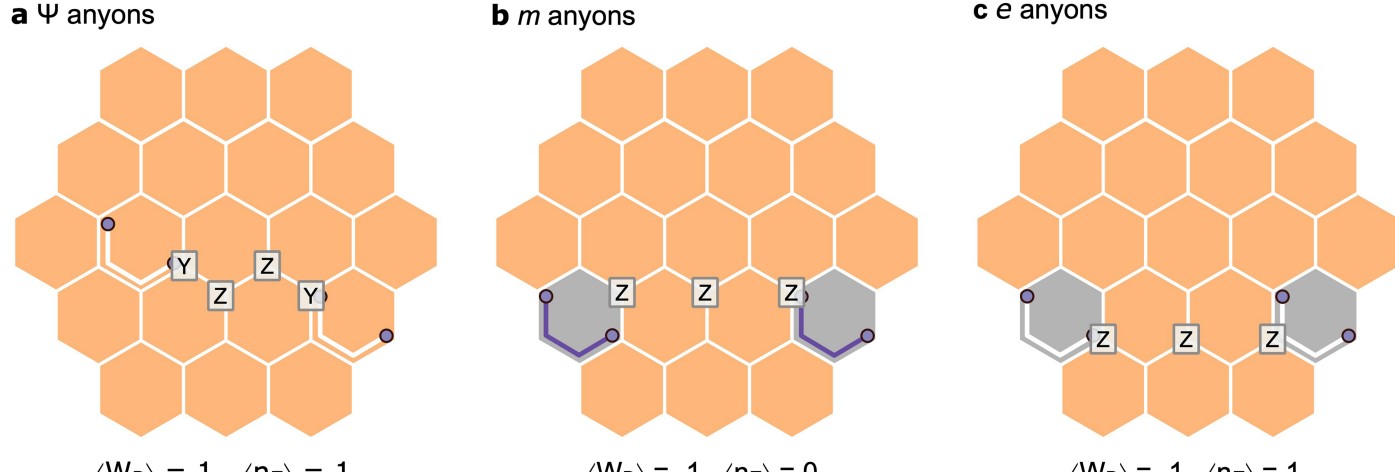

**a** Ψ anyons    **b** *m* anyons    **c** *e* anyons

$\langle W_P \rangle = 1$    $\langle n_F \rangle = 1$      $\langle W_P \rangle = -1$    $\langle n_F \rangle = 0$      $\langle W_P \rangle = -1$    $\langle n_F \rangle = 1$

**Extended Data Fig. 7 | Anyonic excitations. a**, Applying the shown Pauli string leads to the change of occupation of shown bulk fermions resulting in two $\psi$ anyons. **b**, Flipping two fluxes (shown in grey) creates two *m* anyons. **c**, The combination of flipped flux and fermion occupation within a plaquette defines an *e* anyon. The shown Pauli string creates two *e* anyons separated by two plaquettes.

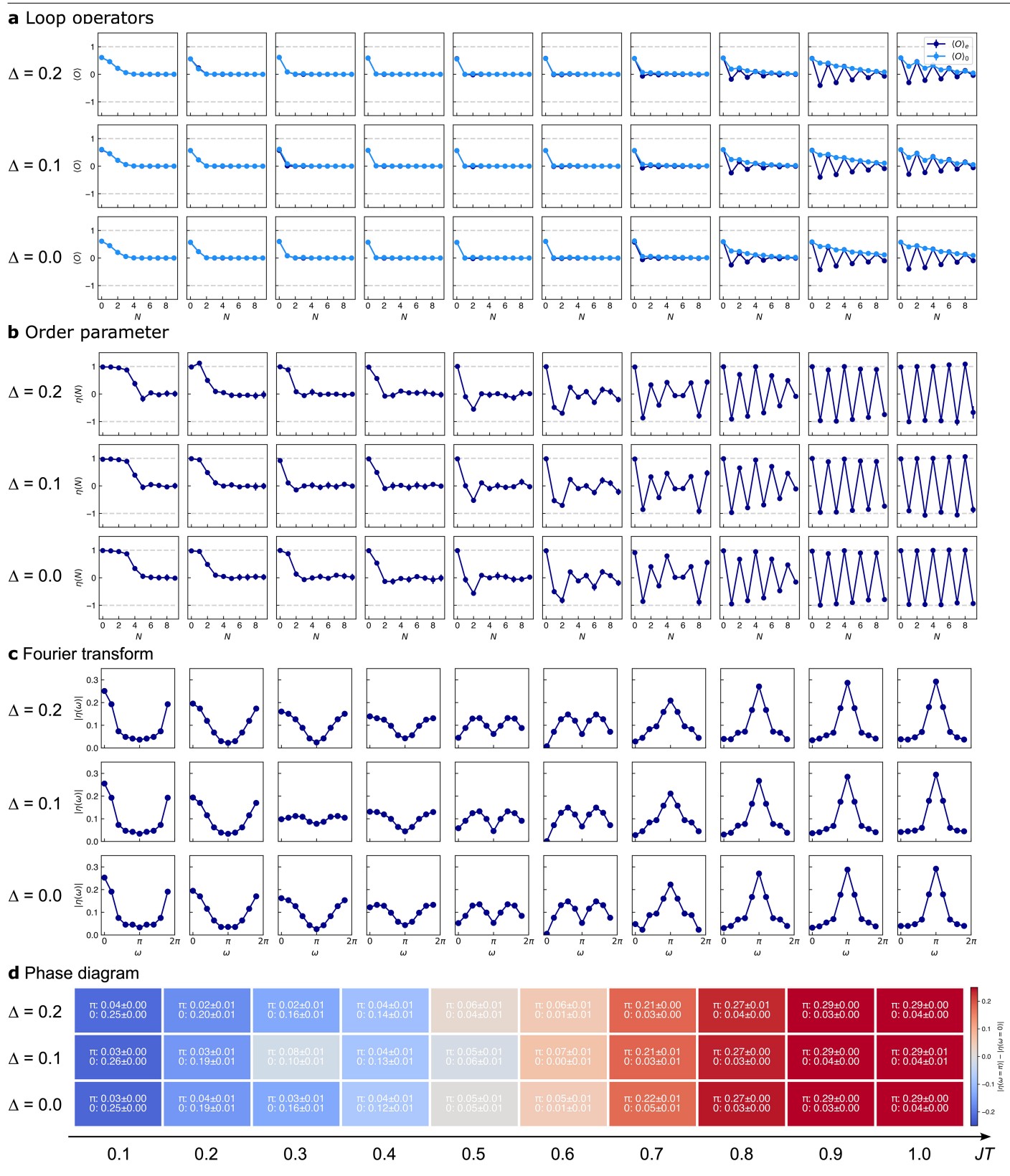

**Extended Data Fig. 8 | Details on the order parameter. a**, In order to extract the phase diagram in Fig. 4e we measure the loop operator over time for an initial state with ($\langle O \rangle_e$) and without ($\langle O \rangle_0$) an $e$ anyon pair. **b**, In the spirit of the Fredenhagen-Marcu operator the order parameter is defined as the ratio of the expectation value of the loop operator of the initial state with the $e$ anyons and the loop operator without the pair. Since both expectation values decay to zero, we add 0.01 to the denominator. In the FTO phase the order parameter shows clear oscillations, which are absent in the Kitaev phase. **c**, Fourier transforming the order parameter reveals a peak at $\pi$ in the FTO phase and a peak at zero in the Kitaev phase. Using the difference of those two values we extract the phase diagram shown in **d**. All uncertainties correspond to $2\sigma$ obtained via jackknife resampling.

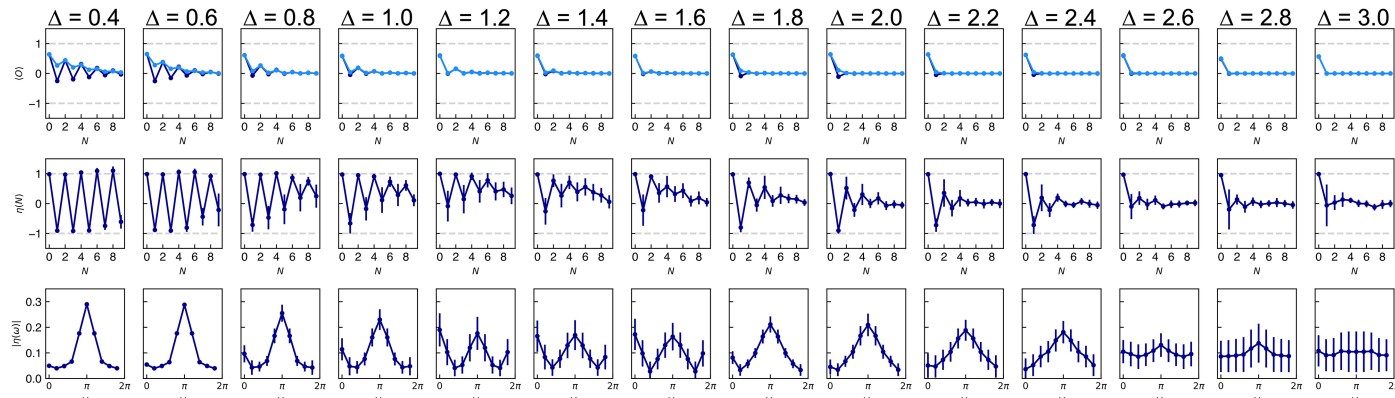

**Extended Data Fig. 9 | Increasing disorder at *JT* = 1.** At *JT* = 1, increasing the disorder strength first leads to an amplitude decay of the oscillations of the order parameter, which eventually disappear at high disorder. This suggests a change from a phase with clear edge dynamics to one where disorder prevents such behavior. Error bars correspond to 2σ obtained via jackknife resampling ($N_{\text{twirling}} = 20$, $N_{\text{shots}} = 10^{6}$, $N_{\text{disorder}} = 50$, 58 qubits).

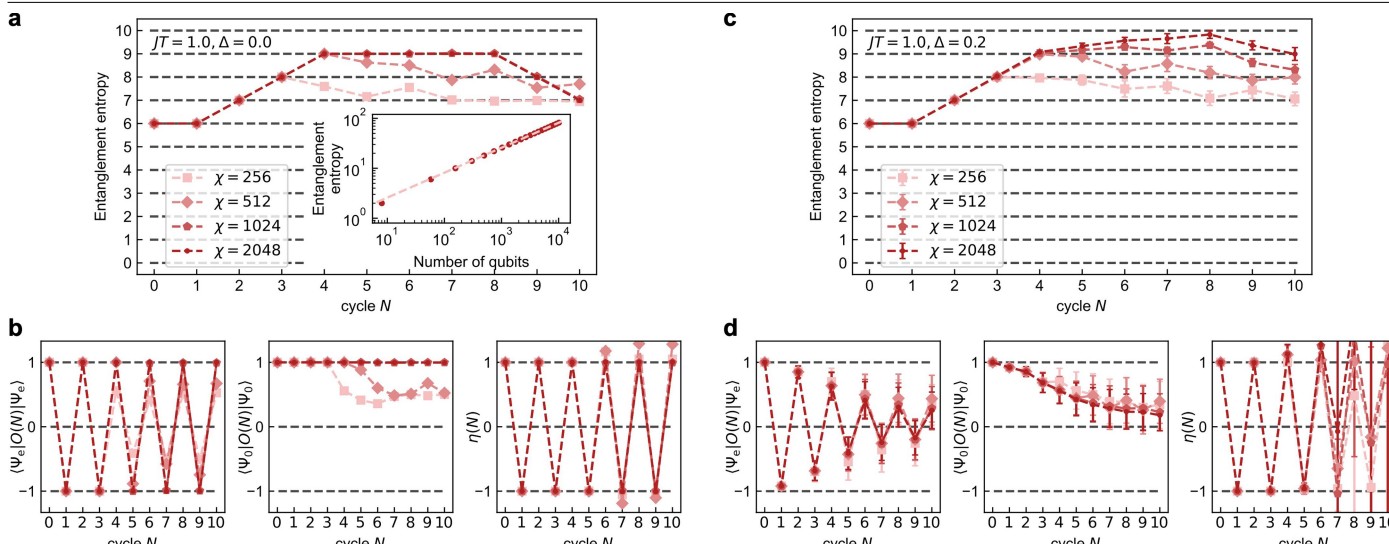

**Extended Data Fig. 10 | MPS simulation of the *e-m* transmutation.** First
we consider the fine-tuned system with *JT* = 1.0 and *Δ* = 0.0. **a**, The maximal
entanglement entropy of all bonds at each time step is shown in units of log(2).
For visual clarity we only show the entanglement entropy for the simulation of
the initial state with an *e* anyon, the results for the initial state without an *e*
anyon are the same. Inset: The red dots show the entanglement entropy of the
initial state when increasing the number of rings in the system, the simulated
system corresponds to three rings or 58 qubits. The light red dashed line shows

the asymptotic scaling $S(N_Q) = \sqrt{2/3\,N_Q}\log(2) + O(1)$, where $N_Q$ is the total
number of qubits in the system. **b**, Left to right: The expectation value of the
loop operator starting from a state with an *e* anyon, without an *e* anyon, and
their ratio which gives the order parameter defined in Eq. (8). **c**, **d**, We show the
same quantities for a system with *JT* = 1.0 and disorder strength *Δ* = 0.2, the data
are averaged over 20 disorder realizations and error bars denote the standard
deviation.