## [Peer Review file · Nature]

Probing Non-Equilibrium Topological Order on a Quantum Processor

Corresponding Author: Professor Frank Pollmann

Version 0:

Reviewer comments:

Referee #1

(Remarks to the Author)

The authors demonstrate on a superconducting quantum processor a dynamical topological state in which a Majorana is chirally transported at the edge and correspondingly e and m anyons are exchanged in the bulk after every cycle. The phenomena of emergent fermions and non-equilibrium dynamics is interesting and timely, illustrating the potential for these devices for fermionic quantum simulation and probing even more complex dynamical behavior. I recommend publication.

I have several questions and comments:

Throughout the paper the authors refer to a "Floquet topologically ordered phase", as a "distinct non-equilibrium phase of matter", but it is not clear to me that it is a sharply defined phase of matter in the thermodynamic limit. Which definition of phase are the authors using? The dynamics is clearly different from a trivial phase, but I'm not sure that is sufficient for rigorously distinguishing two dynamical phases. With disorder, the authors acknowledge that the lack of many-body localization requires one to consider long prethermal time scales. It would be good if the authors can explicitly clarify what they mean by distinct phases in this context.

Regarding the Majorana interferometry, it would be very useful if the authors can elaborate on the decoherence processes contributing to the amplitude and phase errors. For example, I would think that parity flip errors contribute to amplitude decay and parity dephasing contributes to the phase error? Are the parity dephasing errors suppressed by the separation of the two fermions?

On that note, it seems the interferometry experiment could be conducted on a larger device like that used in Fig. 4? The latter involved substantially more qubits because the circuit depths required were less? Would be useful for the authors to clarify the change in system sizes between the experiments.

As the authors hint at the regimes in which classical simulation of the dynamics is difficult, it would be illuminating if they can clarify whether the system sizes and time scales of the current experiment are in such regimes or not (roughly what are the current boundaries of classical simulation).

Referee #2

(Remarks to the Author)

In the work entitled "Probing Non-Equilibrium Topological Order on a Quantum Processor", the authors report the realization of a Floquet topologically ordered (FTO) state in a programmable two-dimensional array of superconducting transmon qubits. Specifically, they perform a digital quantum simulation of the Floquet Kitaev model introduced in PRB 96, 245116 (2017). The gates of the quantum processor implement spin exchange couplings between nearest-neighbours in a

honeycomb lattice array. This spin interaction is locally turned on and off along the lattice bonds in a step-like and periodic manner, thus realizing a Floquet-type evolution. The physics of the model is rationalized by providing an effective description in terms of Z_2 gauge fluxes and Majorana excitations, which can be paired up to form complex fermions.

The manuscript is well-written, with clearly presented experimental data, which is further supported by numerical simulations in the Supplementary Material. The authors convincingly demonstrate the existence of a previously-unobserved Floquet topological phase. In their experiment, they clearly show the emergence of chiral Majorana edge states and their non-abelian nature through interferometric measurements. Even more, they introduce and measure a novel bulk order parameter which reveals the periodic exchange of electric and magnetic anyon excitations in the FTO phase. This allowed them to build-up a pre-thermal phase diagram of their driven system in an extensive region of parameter space. Overall, I judge this to be an important and original piece of work that deserves to be published in Nature.

However, I do have a few questions and comments that should be addressed before publication:

1. In Fig. 3e, the authors attribute the decay of the real-space edge correlator to different mechanisms depending on the value of JT : for $JT=1$, they mention decoherence and coherent hardware errors, while for $JT=0.9$, the decay is said to arise from edge modes spreading into the bulk. The current wording suggests that these contributions can be cleanly disentangled in the experiment. Could the authors clarify whether there is a quantitative or empirical method used to distinguish between decoherence-induced decay and decay due to bulk leakage? If such a distinction is not directly measurable, it would be helpful to revise this statement.
2. One point that could benefit from further clarification is the robustness of the FTO phase under disorder. The results indicate that the phase remains remarkably stable, but it would be helpful to know whether the authors can estimate if there is a critical disorder strength beyond which the phase should become unstable. Additionally, it would be useful to understand whether any experimental constraints prevented exploring larger values of Δ beyond 0.2, and whether this limitation affects the generality of their conclusions.
3. Throughout the manuscript, the data and analysis are presented stroboscopically, i.e., only at the end of each full driving cycle. It would be interesting to know whether access to micromotion—observable quantities at intermediate times within each cycle—is experimentally feasible. For instance, the bulk order parameter is shown to oscillate between ± 1 at stroboscopic times, but one might expect deviations from this behavior at fractional periods (e.g., $T/3$).
4. It is not immediately clear from the main text that the results shown in Fig. 4 were obtained using a different (larger) quantum processor than the one used for the earlier figures. I suggest explicitly noting this change in the main text to avoid potential confusion.

Dear Editor,

Thank you for managing the referee process. We appreciate the constructive feedback of the referees..

Please find our point-by-point response to the referees' comments below.

The formatting for this reply is as follows:

- Direct quotes from reviewers are blue
- Our responses are black
- *Quotes from the manuscript are italicized*
- *Additional text added in response to reviewer comments is highlighted.*

Referee #1 (Remarks to the Author):

The authors demonstrate on a superconducting quantum processor a dynamical topological state in which a Majorana is chirally transported at the edge and correspondingly e and m anyons are exchanged in the bulk after every cycle. The phenomena of emergent fermions and non-equilibrium dynamics is interesting and timely, illustrating the potential for these devices for fermionic quantum simulation and probing even more complex dynamical behavior. I recommend publication.

We thank the reviewer for their time and effort considering our manuscript. We are delighted to hear their recommendation for publication. In the following, we address their questions point by point.

I have several questions and comments:

Throughout the paper the authors refer to a "Floquet topologically ordered phase", as a "distinct non-equilibrium phase of matter", but it is not clear to me that it is a sharply defined phase of matter in the thermodynamic limit. Which definition of phase are the authors using? The dynamics is clearly different from a trivial phase, but I'm not sure that is sufficient for rigorously distinguishing two dynamical phases. With disorder, the authors acknowledge that the lack of many-body localization requires one to consider long prethermal time scales. It

would be good if the authors can explicitly clarify what they mean by distinct phases in this context.

The short answer is that, in the thermodynamic limit, there is a sharp distinction in the absence of interaction (i.e., when the system can be mapped to a quadratic Majorana model) and the stability of the interacting case relies on the stability of many-body localization in 2D. Thus, building on the current understanding that MBL is absent in 2D, the latter case is expected to be only robust to pre-thermal time scales.

Some more details:

- In the absence of interactions, the phases are sharply defined by topological invariants in the thermodynamic limit. In the clean, disorder-free case, a topological invariant can be directly computed from the Floquet band structure [Fulga et al., Phys. Rev. B 99, 235408 (2019)]. Upon introducing disorder that preserves the quadratic Majorana structure, the system enters a Floquet topological Anderson insulating phase, for which the phase can still be sharply characterized, for instance using our bulk invariant based on anyon transmutation.
- In the presence of interactions and sufficiently strong disorder, the transmutation-based invariant remains well-defined. However, whether the interacting phase is truly robust in the thermodynamic limit—or only prethermal—ultimately depends on the stability of many-body localization (MBL). Given the accessible system sizes and time scales in our experiments, we probe the early-time regime and cannot make conclusive statements regarding long-term stability.

Action:

In order to clarify the meaning of the term “phase of matter”, we slightly extended the discussion:

"The FTO, however, represents an extended non-equilibrium phase of matter that remains stable in the thermodynamic limit in the non-interacting case. It is also expected to be protected over long time scales in the presence of interactions, provided there is sufficient disorder to stabilize a pre-thermal many-body localized bulk while retaining FTO [6]."

Regarding the Majorana interferometry, it would be very useful if the authors can elaborate on the decoherence processes contributing to the amplitude and phase errors. For example, I would think that parity flip errors contribute to

amplitude decay and parity dephasing contributes to the phase error? Are the parity dephasing errors suppressed by the separation of the two fermions?

We agree with the referee that a parity flip of the stretched fermion is expected to contribute to amplitude decay, while parity dephasing leads to phase errors. Since the fermion parity can be flipped by a local operator, it is not protected by the spatial separation of the endpoint Majoranas. As we are probing overlaps of full many-body wavefunctions, a sharp distinction between amplitude and phase contributions is not straightforward.

To provide further insights, we present, in the below figure, experimental data (middle) from a two-plaquette setup (left). In (right), we show numerical simulations incorporating local depolarizing noise, amplitude damping on the ancilla to simulate T1 decay, and readout errors. This model qualitatively reproduces the observed amplitude decay and supports the interpretation that local noise mechanisms contribute to amplitude decay.

Action:

We have added this figure along with a detailed description, to the Methods section under "Noisy Simulation."

To further explore the effects of noise on our results, we perform noisy quantum trajectory simulations and compare them to experimental data (Extended Data Fig. e) of a two-plaquette setup, which is shown on the left hand side. Because we utilize randomized compiling, we approximate CZ gate errors as a two-qubit depolarizing channel. In the simulation, we apply different error rates to each pair of qubits, obtained from two-qubit XEB experiments conducted just before the experimental data in Extended Data Fig. e was

collected. We also include amplitude damping on the ancilla during the Floquet cycles to account for T1 decay, assuming a T1 of 25 μ s. Finally, we include readout errors of 0.4% on the $|0\rangle$ state and 2.5% on the $|1\rangle$ state. This error model produces results that are qualitatively consistent with the experimental data: the overlap shows a decaying magnitude, r . Together with the results in Experimental Data Fig. b--d, this agreement suggests the described error model with an additional coherent phase drift on the ancilla qubit (strongly mitigated by dynamical decoupling) reproduces our experimental observations.

On that note, it seems the interferometry experiment could be conducted on a larger device like that used in Fig. 4? The latter involved substantially more qubits because the circuit depths required were less? Would be useful for the authors to clarify the change in system sizes between the experiments.

While the interferometry experiment could, in principle, be extended to larger system sizes, it was conceived as a proof-of-concept demonstration at the fine-tuned point $JT = 1$, specifically to probe the emergence of Majoranas. As this protocol is restricted to $JT=1$, we explore larger system sizes in the e - m transmutation experiment, which allows access to a broader range of parameters.

Action:

To clarify the change in system size, we now explicitly state the system size used in the e - m transmutation experiment and have added the following sentence:

"Probing a system of 58 qubits, the oscillations with period $N=2$ are clearly seen ..."

As the authors hint at the regimes in which classical simulation of the dynamics is difficult, it would be illuminating if they can clarify whether the system sizes and time scales of the current experiment are in such regimes or not (roughly what are the current boundaries of classical simulation).

To assess the hardness of classical simulations, we have attempted to simulate the largest setup, namely the 58-qubit e - m transmutation experiment, using matrix-product states (MPS). This approach becomes inefficient once the entanglement entropy becomes too large, as is the case here. Even for a bond dimension of 2048, we observe that simulation results had not converged after just four Floquet cycles.

Running ten time steps for a single disorder configuration on a CPU with 32 cores took approximately ten days. While this could be improved with optimized parallelization or

more compute resources, it already suggests that obtaining converged results for this system size is at the boundary of classical tractability with MPS methods.

A more rigorous assertion of classical hardness would require testing other classical techniques, such as projected entangled-pair states (PEPS) or Pauli propagation methods, which we have not done. Nevertheless, the observed entanglement growth and simulation times indicate that the current experiment is indeed probing dynamics that are challenging for classical methods to capture.

Action:

We have added data for simulations with bond dimension of 2048 to the figure in Methods (see below, previously we only showed data up to a bond dimension of 1024).

We have further added the following paragraph at the end of the section discussing the hardness of classical simulation:

In general, we see that the entanglement entropy of the state grows linearly in time, initially with a growth rate of $\log(2)$ and followed by a slower rate for the evolution with disorder. While much of the entanglement entropy coming from our choice of initial state could, in principle, be captured by two-dimensional tensor networks, the subsequent growth of the entanglement entropy is generic and expected to eventually violate a two-dimensional area law [67–70]. Thus, we anticipate that classical simulation of this setup will become intractable for any tensor-network-based approach for large systems and late times. Even for the system sizes accessible to us on the quantum processor, we cannot achieve convergence even with the largest bond dimension $\chi=2048$ that we have simulated. Further, since at every Floquet cycle the circuit now introduces both non-Clifford operations and interacting fermionic

gates, we expect the simulation to also be hard with techniques building on the Clifford or free-fermion structure. Note, however, that we have not explicitly checked these or other methods (e.g. projected entangled-pair states, isometric tensor-network states, neural network states, ...) directly.

Referee #2 (Remarks to the Author):

In the work entitled "Probing Non-Equilibrium Topological Order on a Quantum Processor", the authors report the realization of a Floquet topologically ordered (FTO) state in a programmable two-dimensional array of superconducting transmon qubits. Specifically, they perform a digital quantum simulation of the Floquet Kitaev model introduced in PRB 96, 245116 (2017). The gates of the quantum processor implement spin exchange couplings between nearest-neighbours in a honeycomb lattice array. This spin interaction is locally turned on and off along the lattice bonds in a step-like and periodic manner, thus realizing a Floquet-type evolution. The physics of the model is rationalized by providing an effective description in terms of Z_2 gauge fluxes and Majorana excitations, which can be paired up to form complex fermions.

The manuscript is well-written, with clearly presented experimental data, which is further supported by numerical simulations in the Supplementary Material. The authors convincingly demonstrate the existence of a previously-unobserved Floquet topological phase. In their experiment, they clearly show the emergence of chiral Majorana edge states and their non-abelian nature through interferometric measurements. Even more, they introduce and measure a novel bulk order parameter which reveals the periodic exchange of electric and magnetic anyon excitations in the FTO phase. This allowed them to build-up a pre-thermal phase diagram of their driven system in an extensive region of parameter space. Overall, I judge this to be an important and original piece of work that deserves to be published in Nature.

We thank the reviewer for their thoughtful and encouraging assessment of our manuscript, and we are pleased that they view the work as "an important and original piece of work" and recommend it for publication. We also appreciate the constructive feedback provided and address each of the reviewer's comments point by point.

However, I do have a few questions and comments that should be addressed before publication:

1. In Fig. 3e, the authors attribute the decay of the real-space edge correlator to different mechanisms depending on the value of JT : for $JT=1$, they mention decoherence and coherent hardware errors, while for $JT=0.9$, the decay is said to arise from edge modes spreading into the bulk. The current wording suggests that these contributions can be cleanly disentangled in the experiment. Could the authors clarify whether there is a quantitative or empirical method used to distinguish between decoherence-induced decay and decay due to bulk leakage? If such a distinction is not directly measurable, it would be helpful to revise this statement.

We assume that hardware-induced errors are present across all values of JT . In the ideal case at $JT = 1$, the edge mode is perfectly localized, and the diagonal elements of the real-space spectral function are expected to remain pinned at 1.0 for all times. For $JT = 0.9$, numerical simulations show that the edge state extends into the bulk, reducing the diagonal element to 0.94 after 10 Floquet cycles.

In the experiment, hardware errors generally lead to signal decay for all JT values. At $JT = 1$, we measure an initial diagonal value of 0.90 ± 0.003 following state preparation, which decays to 0.14 ± 0.003 after 10 Floquet cycles. For $JT = 0.9$, the signal was 0.89 ± 0.003 after state preparation and further decays to 0.02 ± 0.006 over the same duration of 10 cycles. We attribute this additional suppression relative to the $JT = 1$ case to increased leakage into bulk modes.

Action:

We thank the referee for pointing out the ambiguous phrasing and have revised the main text for clarity. Specifically,

"The observed decay at $JT = 1$ is due to the combination of coherent errors and decoherence in the device, which affects all other data sets as well."

2. One point that could benefit from further clarification is the robustness of the FTO phase under disorder. The results indicate that the phase remains remarkably stable, but it would be helpful to know whether the authors can estimate if there is a critical disorder strength beyond which the phase should become unstable. Additionally, it would be useful to understand whether any experimental constraints prevented exploring larger values of Δ beyond 0.2, and whether this limitation affects the generality of their conclusions.

In Ref. [Titum *et al.*, PRX 6, 021013 (2016)], a bosonic analogue of the driven Kitaev model was investigated in the presence of disorder. In this so-called anomalous Floquet-Anderson insulator, bosons reside on a square lattice with hopping terms that are alternately activated along the bonds of each plaquette. Using level-spacing statistics, the authors predict a transition to a fully localized phase at a critical disorder strength.

We expect similar behavior in the Floquet Kitaev model: namely, a transition to a localized phase at sufficiently strong disorder.

To explore this further, we have taken additional experimental data at $JT=1$ and increasing values of Δ , see the figure below. The order parameter exhibits oscillations up to $\Delta \approx 2.4$, with a gradual reduction in amplitude. For larger values of Δ , these oscillations seem to vanish.

Due to resource constraints, we are limited to averaging over a finite number of disorder realizations and twirling sets. Using jackknife resampling, we are able to extract results using 50 disorder instances and 20 twirling sets. However, at higher disorder strengths, the limited samples prevent us from making definitive statements.

Nonetheless, these preliminary observations highlight a promising direction for future experiments aimed at probing the interplay between Floquet topological order and localization physics.

Action:

We have added the following figure showing an additional data parameter scan and explanations to the Methods section.

To investigate the impact of the combination of interactions and disorder, we performed measurements at $JT=1$ with increasing disorder strength Δ , see Extended Data Fig. 10. For small disorder (roughly $\Delta \leq 2.4$), the order parameter shows oscillatory behavior, indicative of the FTO phase, though the amplitude gradually decreases as Δ

increases. At higher disorder strengths, these oscillations become suppressed and eventually disappear. The observed behavior suggests a crossover or transition towards a localized phase at strong disorder. However, due to limited experimental resources, we averaged only over 50 disorder realizations for each data point. As a result, we cannot make definitive claims about the nature of the transition.

Furthermore, we added a sentence to the main text referring to these datasets.:

Additional datasets for higher disorder at $JT=1$ are presented in the Methods section[42].

In addition we have added a section on "Data analysis" to the Methods section explaining the jackknife resampling method. We have now applied jackknife resampling to all datasets to keep error analysis consistent for all presented data .

For all datasets, we apply jackknife resampling prior to further analysis. The raw data for each experiment consists of 20 twirling sets, and each twirling set contains many measured data points. To estimate statistical uncertainties, we construct jackknife samples by systematically leaving out one entire twirling set at a time.

More precisely, let θ_i denote the average for the i -th twirling set. Then the i -th jackknife sample j_i is given by:

$$j_i = \frac{1}{N_{\text{twirling}} - 1} \sum_{k \neq i}^{N_{\text{twirling}}} \theta_k. \quad (14)$$

This procedure yields $N_{\text{twirling}} = 20$ jackknife samples j_i . When analyzing a quantity f that depends on multiple measured variables, we construct jackknife samples for each variable on its own. Suppose our desired analysis function is a multivariate function f , then for each jackknife index i we have a set of jackknife samples (j_i, l_i, \dots) corresponding to all variables. Applying f to these jackknife samples yields 20 samples:

$$F_i = f(j_i, l_i, \dots). \quad (15)$$

Finally, the jackknife mean and standard error are given by:

$$\bar{F} = \frac{1}{N_{\text{twirling}}} \sum_{i=1}^{N_{\text{twirling}}} F_i \quad (16)$$

$$\sigma = \sqrt{\frac{N_{\text{twirling}}-1}{N_{\text{twirling}}} \sum_{i=1}^{N_{\text{twirling}}} (F_i - \bar{F})^2}. \quad (17)$$

In all figures error bars denote $\pm 2\sigma$ as a measure of spread.

As a concrete example, in the braiding experiment we measure the real and imaginary parts of the complex amplitude separately. Jackknife resampling is applied to each dataset independently, yielding 20 jackknife estimates for the real part and 20 for the imaginary part. The corresponding real and imaginary sets are then paired up for each jackknife index to reconstruct the full complex amplitude. From these pairs we compute the radial and angular components, and finally estimate the mean and standard error for each observable across this jackknife ensemble.

3. Throughout the manuscript, the data and analysis are presented stroboscopically, i.e., only at the end of each full driving cycle. It would be interesting to know whether access to micromotion—observable quantities at intermediate times within each cycle—is experimentally feasible. For instance, the bulk order parameter is shown to oscillate between ± 1 at stroboscopic times, but one might expect deviations from this behavior at fractional periods (e.g., $T/3$).

Accessing stroboscopic time steps within a single driving cycle is experimentally feasible. For instance, in Fig. 1, additional measurements taken after the application of the X (i.e., $T/3$) and Y (i.e., $2T/3$) pulses would provide information about the fermion positions at those intermediate times. However, the transmutation of e -anyons into m -anyons is only complete after a full driving cycle comprising X , Y , and Z steps.

The definition of the e -loop operator relies on the representation of e -anyons immediately prior to the Z pulse. As such, measurements of this operator at intermediate times would not correspond to the intended e -loop and would not capture the desired topological information.

4. It is not immediately clear from the main text that the results shown in Fig. 4 were obtained using a different (larger) quantum processor than the one used for the earlier figures. I suggest explicitly noting this change in the main text to avoid potential confusion.

We appreciate this comment and have adjusted the main text accordingly. Namely we now also specify the system size of the e-m transmutation experiment in the main text.

Action:

"Probing a system of 58 qubits, the oscillations with period $N=2$ are clearly seen ..."